# Separable gain control of ongoing and evoked activity in the visual cortex by serotonergic input

**Zohre Azimi**[1,2], **Ruxandra Barzan**[1,2], **Katharina Spoida**[3], **Tatjana Surdin**[3], **Patric Wollenweber**[3], **Melanie D Mark**[3], **Stefan Herlitze**[3], **Dirk Jancke**[1,2]*

[1]Optical Imaging Group, Institut für Neuroinformatik, Ruhr University Bochum, Bochum, Germany; [2]International Graduate School of Neuroscience (IGSN), Ruhr University Bochum, Bochum, Germany; [3]Department of General Zoology and Neurobiology, Ruhr University Bochum, Bochum, Germany

**Abstract** Controlling gain of cortical activity is essential to modulate weights between internal ongoing communication and external sensory drive. Here, we show that serotonergic input has separable suppressive effects on the gain of ongoing and evoked visual activity. We combined optogenetic stimulation of the dorsal raphe nucleus (DRN) with wide-field calcium imaging, extracellular recordings, and iontophoresis of serotonin (5-HT) receptor antagonists in the mouse visual cortex. 5-HT1A receptors promote divisive suppression of spontaneous activity, while 5-HT2A receptors act divisively on visual response gain and largely account for normalization of population responses over a range of visual contrasts in awake and anesthetized states. Thus, 5-HT input provides balanced but distinct suppressive effects on ongoing and evoked activity components across neuronal populations. Imbalanced 5-HT1A/2A activation, either through receptor-specific drug intake, genetically predisposed irregular 5-HT receptor density, or change in sensory bombardment may enhance internal broadcasts and reduce sensory drive and vice versa.

*For correspondence:
dirk.jancke@rub.de

Competing interests: The authors declare that no competing interests exist.

## Introduction

Brain networks manifest a continuous interplay between internal ongoing activity and stimulus-driven (evoked) responses to form perception. Numerous studies have shown that cortical states modulate ongoing activity and its interaction with sensory-evoked input (*Arieli et al., 1996*; *Deneux and Grinvald, 2016*; *Ferezou and Deneux, 2017*; *Ferezou et al., 2007*; *Fiser et al., 2010*; *Fox et al., 2006*; *Haider et al., 2007*; *He, 2013*; *Kasanetz et al., 2002*; *Kenet et al., 2003*; *Luczak et al., 2009*; *Markram et al., 1998*; *McGinley et al., 2015*; *Niell and Stryker, 2010*; *Raichle, 2015*; *Schölvinck et al., 2015*; *Shimaoka et al., 2019*) that is further controlled by various gain mechanisms (*Carandini and Heeger, 2012*). Neuromodulators that affect the state of cortical networks (*Marder, 2012*), however, can alter the balance between ongoing (partly top-down) and evoked (largely bottom-up) activity. Therefore, they can crucially influence the formation of perceptual events (*Cassidy et al., 2018*; *Pinto et al., 2013*; *Yu and Dayan, 2002*) and higher cognitive functions (*Doya, 2008*; *Hesselmann et al., 2008*; *Hurley et al., 2004*; *Sadaghiani et al., 2010*). One prominent neuromodulator involved in the modulation of the cortical state is serotonin (5-hydroxytryptamine; 5-HT) (*Rapport et al., 1948*), which is mainly released from 5-HT neurons in the dorsal raphe (DR) and median raphe (MR) nuclei (*Dahlström and Fuxe, 1964*; *Descarries et al., 1982*; *Ishimura et al., 1988*; *Mengod et al., 2006*). The serotonergic system comprises widespread projections to all cortical and subcortical areas (*Hale and Lowry, 2011*; *Jacobs and Azmitia, 1992*; *Mengod et al., 2006*; *Vertes and Linley, 2007*; *Waterhouse et al., 1986*) with different 5-HT receptors (inhibitory or depolarizing) co-distributed across different cortical cell types (*Hannon and*

*Hoyer, 2008*; *Leysen, 2004*; *Santana et al., 2004*). Altogether, this makes 5-HT a likely candidate for the contribution to fine-tuned scaling of both ongoing and evoked activity (*Shimegi et al., 2016*) as well as their integration across brain networks (*Berger et al., 2009*; *Conio et al., 2019*; *Lesch and Waider, 2012*).

In the visual cortex, pioneering studies using electrical stimulation of the dorsal raphe nucleus (DRN) revealed its general modulatory influence on evoked cortical responses (*Gasanov et al., 1989*; *Moyanova and Dimov, 1986*). Cortical application of 5-HT or 5-HT receptor agonists via microiontophoresis and single-cell recordings in vivo showed either suppressive or facilitative effects (*Krnjević and Phillis, 1963*; *Reader, 1978*; *Watakabe et al., 2009*; *Waterhouse et al., 1990*). Ionto-phoretic application of 5-HT into the visual cortex of awake monkeys in a recent study demonstrated that at the population level, however, 5-HT decreases mainly the gain of evoked responses without effecting spontaneous activity (*Seillier et al., 2017*). Such a distinct 5-HT effect on response gain was recently attributed to selective activation of 5-HT2A receptors (*Zhang and Stackman, 2015*) via several different approaches: after subcutaneous injection of a hallucinogenic 5-HT2A receptor ago-nist in mice (*Michaiel et al., 2019*), through whole-brain modelling (*Deco et al., 2018*) of human 5-HT2A receptor density and stimulation with lysergic acid diethylamide (LSD), and by direct and spe-cific optogenetic activation of 5-HT2A receptors in the mouse visual cortex (*Eickelbeck et al., 2019*). Intriguingly, using optogenetic stimulation of 5-HT neurons in the mouse DRN (*Dugué et al., 2014*; *Kapoor et al., 2016*; *Matias et al., 2017*), only one study in olfactory cortex has shown so far a 5-HT-induced (divisive) gain control of spontaneous firing, without any effect on the gain of stimu-lus-driven population responses (*Lottem et al., 2016*). This suggests that DRN activation could sepa-rately reduce the weight of ongoing cortical activity relative to evoked activity (*Lottem et al., 2016*), thereby possibly changing the balance of integration between internal priors (*Berkes et al., 2011*; *Fiser et al., 2010*) and external sensory input (*Lottem et al., 2016*).

In this study, we employed optogenetic stimulation of 5-HT neurons in the DRN to investigate 5-HT-induced effects on ongoing and evoked population activity in the mouse primary visual cortex (V1) using wide-field $Ca^{2+}$ imaging of RCaMP signal complemented with recordings of multi-unit spiking activity. Our results provide evidence for 5-HT-induced separable suppression of evoked and ongoing activity, affecting the gain of both these components in a divisive manner. Concurrent ion-tophoretic application of specific antagonists of 5-HT1A and 5-HT2A receptors (*Dyck and Cynader, 1993*; *Jakab and Goldman-Rakic, 1998*; *Leysen, 2004*; *Riga et al., 2016*; *Shukla et al., 2014*) sug-gest a distinct role of these receptors in regulating suppression levels of spontaneous and evoked population activity, respectively (*Azimi et al., 2018*).

Finally, we investigated how the divisive nature of 5-HT modulation affects evoked population responses to varying input intensities and integration with ongoing activity (*Deco et al., 2015*). Addressing this question is important because it allows estimating the extent of 5-HT-induced scal-ing of activity preserving the information content, as formalized by a model of divisive normalization (*Carandini and Heeger, 2012*). We found that normalization of responses in the anesthetized state is achieved with additive contribution of ongoing activity, indicating integration of spontaneous and evoked activity with increased weight of internal priors. In awake mice, normalization is largely inde-pendent of suppression in ongoing activity. This suggests that 5-HT provides a discrete divisive gain control of spontaneous ongoing and evoked visual activity, while preserving information content and regulating the balance between these components depending on the cortical state.

## Results

To trigger the activation of 5-HT neurons in the DRN with precise timing, we used a transgenic ePet-Cre mouse line (*Scott et al., 2005*), which allows expression of Channelrhodopsin2 (ChR2) in 5-HT neurons by Cre-dependent expression of double-floxed adeno-associated virus (AAV, see Materials and methods). This enabled real-time activation of 5-HT neurons via photostimulation (*Li et al., 2005*; *Figure 1*) in vivo (see *Figure 1—figure supplement 1* for an example of extracellular recordings in the DRN). In order to simultaneously record activity of a large number of neurons across V1, we employed wide-field optical imaging of $Ca^{2+}$ signals, shown to reflect suprathreshold population activity across upper cortical layers (*Kim et al., 2016*; *Lütcke et al., 2013*; *Lütcke et al., 2010*; *Wallace et al., 2008*; *Xiao et al., 2017*). Specifically, we used the red-shifted fluorescent probe RCaMP (*Akerboom et al., 2013*; *Dana et al., 2016*; *Figure 1a*) to minimize interference with

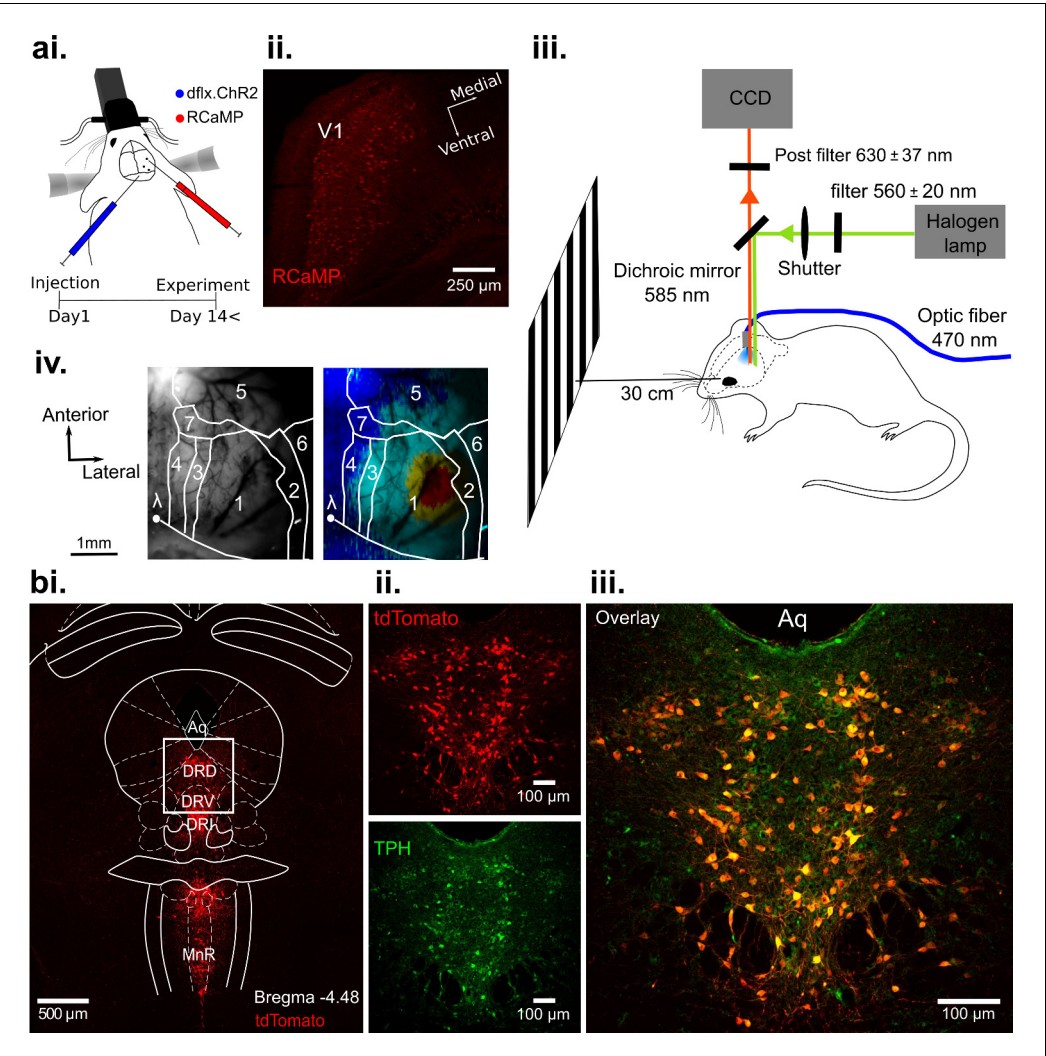

**Figure 1.** Optogenetic activation of DRN 5-HT neurons and concurrent RCaMP imaging in V1 in vivo. (**ai**) Viral injection of RCaMP in V1 and adjacent cortical areas and injection of ChR2-mCherry into DRN of ePet-Cre mice. (**aii**) Coronal section of V1, showing expression of RCaMP. (**aiii**) Schematic of the experimental setup. DRN photostimulation was performed via an implanted optic fiber and wide-field imaging was achieved through the thinned skull. Stimuli were displayed on a monitor at 30 cm distance to the eye contralateral to the recorded hemisphere. Animals were anesthetized and head-fixed. (**aiv**) Vascular pattern (left) of the imaged cortical region showing activation across V1 and neighboring visual areas after visual stimulation (right). 1: V1; 2: V2$_{Lateral}$; 3: V2$_{Medio-Lateral}$; 4: V2$_{Medio-Medial}$; 5: somatosensory cortex; 6: auditory cortex; 7: PtA (parietal association area). (**bi**) Coronal section at the DRN injection site after expression of Cre-dependent tdTomato (red). Most subnuclei including dorsal (DRD), ventral (DRV), and interfascicular (DRI) parts of the DRN, and the median raphe nucleus (MnR) show fluorescent labeling of serotonergic cells. (**bii-iii**) Magnified view of the area outlined in (**bi**). Labeling with fluorescent reporter tdTomato (**bii**, top), antibody labeling against tryptophan hydroxylase (TPH, (**bii**), bottom), and their co-localization (**biii**); Aq: aquaduct.

The online version of this article includes the following figure supplement(s) for figure 1:

**Figure supplement 1.** Example extracellular MUA recording in the DRN during photostimulation in an ePet-Cre mouse.

the blue light employed to activate serotonergic neurons in the DRN (*Figure 1b*) and to reduce light scattering caused by hemodynamic signals in comparison to GCaMP.

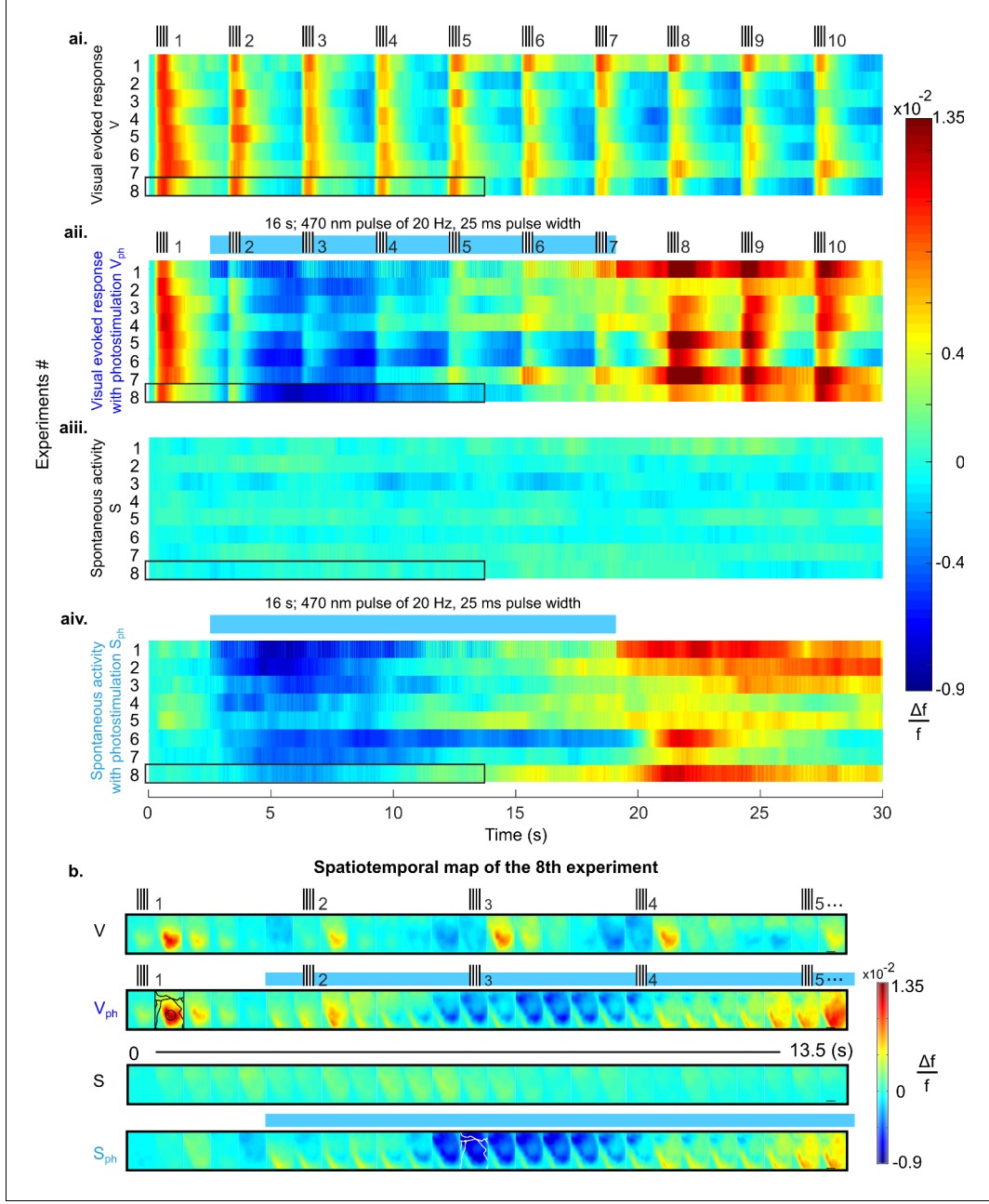

**Figure 2.** Suppression of cortical population activity after photostimulation of 5-HT neurons in the DRN. (**ai-aiv**) Spatial averages of changes in fluorescence (Δf/f, cf. colorbars) across V1 of 8 different mice in response to four experimental conditions. (**ai**) Repeated visual stimulation with drifting gratings (V, icons on top show on/offsets of visual stimulus). (**aii**) Visual stimulation and concurrent photostimulation of the DRN ($V_{ph}$, blue bar on top marks photostimulation time). (**aiii**) Spontaneous activity (S). (**aiv**) Spontaneous activity and concurrent photostimulation of the DRN ($S_{ph}$). (**b**) Example spatiotemporal map of experiment #8 (encircled black in **ai-aiv**) depicting recording across the cortical surface before spatial averaging (image frames were 500 ms time binned) over 13.5 s. Horizontal black lines in right frames delineate 1 mm. Conditions are denoted as in **a**. Different cortical areas (as specified in *Figure 1aiv*, caption) are outlined black and white (see frames within rows two and four, respectively). Small signal increases at lower left corners of image frames simultaneous with onset of photostimulation indicate artifacts due to partial interference with fluorescent signals.

## Photostimulation of 5-HT neurons in the DRN suppresses neuronal responses in primary visual cortex (V1)

As control conditions we recorded evoked responses to visual stimuli (vertical grating with 100% contrast, presented 10 times at intervals of 3 s with 200 ms duration) and spontaneous activity over a total of 30 s (*Figure 2ai*, [V] and *Figure 2aiii* [S], respectively). In addition, we recorded the activity during these conditions with photostimulation (16 s train of 470 nm light pulses at 20 Hz and 50% duty cycle, marked as blue bars in *Figure 2aii*, [$V_{ph}$] and *Figure 2aiv*, [$S_{ph}$]) to activate 5-HT neurons in the DRN. For each condition, *Figure 2a* shows the time course of RCaMP signals as mean spatial averages across V1 (25–50 trials per condition) derived from eight animals under anesthesia. Under control conditions without DRN photostimulation, activation over V1 shows each stimulus occurrence as a rapid ramp-up of the RCaMP signal followed by a slower decay towards baseline level (*Figure 2ai*). In contrast, after the onset of photostimulation, a strong suppression of evoked visual responses is observed, succeeded by a subsequent increase of the $Ca^{2+}$ signal above pre-stimulation levels after cessation of photostimulation (*Figure 2aii*).

Importantly, 5-HT-mediated suppression of cortical activity is present without external visual input; that is, following photostimulation, ongoing cortical activity rapidly declines below baseline levels (*Figure 2aiv*). This suppression of spontaneous activity includes cortical areas beyond V1 (white contours in *Figure 2b*, 4th row; for assignment of cortical areas, see *Figure 1aiv*), suggesting 5-HT affects spontaneous drive through widespread ascending projections from the DRN across the entire cortex (*Hale and Lowry, 2011*).

## Suppression of ongoing activity

We next quantified the effects of DRN photostimulation on spontaneous cortical activity. The traces in *Figure 3ai* depict spatial averages of $Ca^{2+}$ signals over V1. A comparison between spontaneous activity under control conditions (black stippled lines) and upon DRN photostimulation ($S_{ph}$, light blue line) reveals a significant suppression ($-0.10\pm0.02$, n=8 animals, p=0.039; one-sample t test; *Figure 3aii*, left bar). The suppression is significant 680 ms (8 animals, p=0.04; paired t test with permutation correction) after the onset of DRN photostimulation, then it reaches a local minimum followed by an increase in activity (*Figure 3ai*). This later elevation in the RCaMP signal could indicate an increase of intracellular $Ca^{2+}$ levels (*Eickelbeck et al., 2019*) associated with the 5-HT-receptor-mediated activation of the Gq/11 pathway (*Millan et al., 2008*) and concomitant activation of store-operated channels (*Celada et al., 2013*). By restricting the averaging window to the period where suppression of the RCaMP signal is significant (as marked with red rectangle in *Figure 3ai*), the amount of suppression increased to $-0.33\pm0.02$ (8 animals, p<0.001, one-sample t test; *Figure 3aii*, right bar).

To investigate whether the observed suppression is indeed sustained during photostimulation and whether it reflects a decrease in spiking output of V1, we recorded multi-unit activity (MUA), n=9 animals, 104 MUAs, using the same experimental paradigm as in the wide-field RCaMP imaging. Similar to RCaMP imaging, spontaneous activity revealed by MUAs is significantly (below two times standard deviation of pre-photostimulation activity) suppressed after $750\pm184$ ms of DRN photostimulation in comparison to baseline spontaneous firing (*Figure 3bi*, light blue and stippled black line, respectively) when averaged over the early phase (red rectangle in *Figure 3ai and 3bi*) of suppression ($-0.48\pm0.07$, n=9 animals, p<0.001, one-sample t test; *Figure 3bii*, right bar). Note that in contrast to the RCaMP signal, spiking activity is devoid of a subsequent elevation. Instead, the suppression remains highly significant throughout photostimulation ($-0.43\pm0.08$; n=9 animals, p<0.001; one-sample t test; *Figure 3bii*, left bar). This further suggests that the observed rise of the RCaMP signal may represent intracellular accumulation of $Ca^{2+}$, rather than increase in spiking activity. The assumption is also supported by subsequent 5-HT receptor blocking experiments (see below). Altogether, it can be inferred that augmented activity of 5-HT neurons via DRN photostimulation results in suppression of spontaneous activity in the visual cortex. Additional analysis using linear regression shows that this suppression is divisive (*Figure 4*).

## Suppression of visually evoked activity

Next, we compared the conditions with visual input (V and $V_{ph}$). While visually evoked responses of control conditions are characterized by repeated increase of the $Ca^{2+}$ signal with a small adaptive

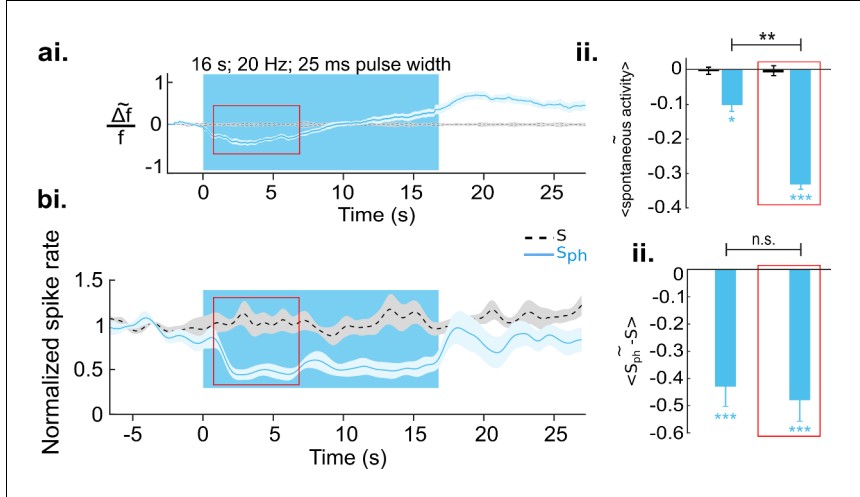

**Figure 3.** Increased serotonergic input suppresses spontaneous cortical activity. (**ai**) Ca$^{2+}$ imaging of spontaneous activity. Traces were derived after spatial averaging and depict mean across eight animals, each experiment comprising 25–50 trials (light blue area marks SEM). Black stippled lines indicate spontaneous activity without photostimulation (S). Blue solid line shows spontaneous activity with concurrent DRN photostimulation (S$_{ph}$). Blue background marks time window of photostimulation. Red rectangle encircles a time window in which S$_{ph}$ is significantly lower than S (see main text). (**aii**) Average of S (black bar) and S$_{ph}$ (light blue bar) over the entire time of photostimulation and over the time window marked by the red rectangle in **ai**. (**bi**) MUA recordings, same conditions as shown in **a**. Data are based on 104 multi-units over 25 different recordings (10–20 trials) in nine animals at cortical depths between 250 and 750 μm. (**bii**) Similar quantification as in **aii** (for details see Methods). All values in the panels are mean ± SEM. Color legend in **bi** applies to all panels. ***p<0.001, **p<0.01, and *p<0.05, one-sample and paired t test. For a single example of MUA recording see *Figure 3—figure supplement 1*.

The online version of this article includes the following figure supplement(s) for figure 3:

**Figure supplement 1.** Multi-unit recording example of spontaneous activity without and with DRN photostimulation.

decrease in the response amplitudes over time (*Figure 5ai and 5bii*, black trace), the amplitude of activity declines toward negative values in the presence of DRN photostimulation (*Figure 5aii*, dark blue trace). Note that both the amplitude of the evoked responses and their baseline are reduced. To assess how much of the suppression in amplitude is due to the suppression of its baseline level, we subtracted image frames obtained under S$_{ph}$ conditions from those under V$_{ph}$ conditions (pixel-wise and across single trials). The outcome (*Figure 5aii*, gray trace) is the photostimulation-induced suppression of the evoked response independent of suppression in spontaneous activity, that we refer to as the evoked component.

To quantify these observations, three measurements are depicted in *Figure 5b*. (see Methods for details). The first measurement is the amplitude of evoked activity, defined as an average over the time window $w_1$ (*Figure 5ai* inset). Following DRN photostimulation, amplitudes of evoked responses (*Figure 5bi* dark blue curve) and evoked component (*Figure 5bi* gray curve) are significantly lower than those of the control condition (*Figure 5bi* black curve; p<0.05; paired t test with permutation correction). In the early part of the photostimulation window (stimulus #2–4), however, amplitudes of evoked responses are also significantly below amplitudes of the evoked component (p<0.05; paired t test with permutation correction). Using a second measurement, that is, the baseline of the traces (averaged over a 200 ms time window before the onset of each visual stimulus ($w_2$, *Figure 5ai* inset) indicates that this difference arises from additional suppression seen in spontaneous activity (*Figure 3ai*). Indeed, the time course of the baseline in the evoked responses during photostimulation is analogous to spontaneous activity during photostimulation (*Figure 5bii*, cf. dark blue and light blue traces, respectively); both are characterized by an initial suppression followed by the previously referred rise in RCaMP signal. We, therefore, define the baseline of the evoked response as the baseline component and it is approximated via spontaneous activity.

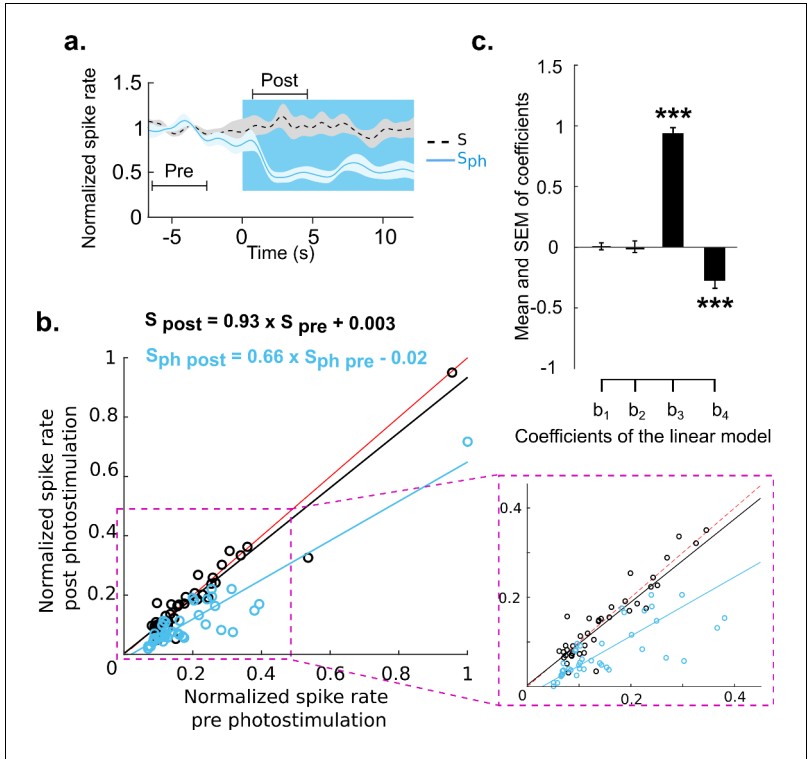

**Figure 4.** Divisive suppression of spontaneous activity following photostimulation of the DRN. (a) Average of spontaneous activity across MUA recordings (41, MUA with pre photostimulation firing rates between 1 and 20 Hz). Pre- and post-photostimulation time windows used for analysis are marked (the length of each window is 4 s). Both traces are normalized to pre-stimulus firing rate, for details see Materials and methods. (b) Comparison between the mean of $S_{ph}$ pre and $S_{ph}$ post photostimulation for each unit (light blue circles) and the same comparison for S (black circles); data points normalized over all units. Solid lines represent the linear regression for $S_{ph}$ (blue) and S (black) and red lines show x=y. Equations of the regressor lines are shown with corresponding colors. The boxed region is a zoom in of the outlined area shown in left panel. (c) Bars show the regression coefficients (mean ± SEM); ***p<0.001, one-sample t test.

In order to verify that the suppression of the evoked component is independent of the suppression of spontaneous activity, the magnitude of responses (i.e. response gain, calculated as absolute change of the amplitude from baseline) is calculated as a third measurement (*Figure 5biii*). The suppression of magnitude is highly significant as compared to control (*Figure 5biii*, blue and black curves, respectively; p<0.05, paired t test with permutation correction). Moreover, in the photostimulated condition the magnitude of the evoked response resembles the evoked component (*Figure 5biii*, cf. overlapping blue and gray curves, respectively; across all stimuli, #1:10, p>0.21, paired t test with permutation correction), confirming that the suppression in response magnitude is separable from suppression of spontaneous activity.

Extracellular recordings of MUA in V1 substantiate the observed suppressive effects on evoked responses and their independence of baseline suppression at the spiking level (*Figure 5—figure supplement 1*). *Figure 5—figure supplement 1b* shows the quantification of the spike recordings using similar analysis as applied to the Ca²⁺ imaging data and *Figure 5—figure supplement 1c* shows an example recording of MUA.

*Figure 5c* summarizes a comparison between Ca²⁺ imaging data and MUA data using average of amplitude and magnitude values during the period of photostimulation (stimulus #2–7) after subtraction of controls. Evoked responses (blue) and evoked component (gray), obtained with Ca²⁺ imaging and MUA indicate a highly significant reduction in amplitude (*Figure 5ci*, Ca²⁺ imaging data: −0.450.02 [blue] and −0.39±0.02 [gray], n=8 animals, p<0.001, one-sample t test; MUA data: −0.55±0.04 [blue] and −0.41±0.04 [gray], n=9 animals, p<0.001, one-sample t test). The smaller suppression in the amplitude of the Ca²⁺ signal compared to MUA data reflects that the chosen

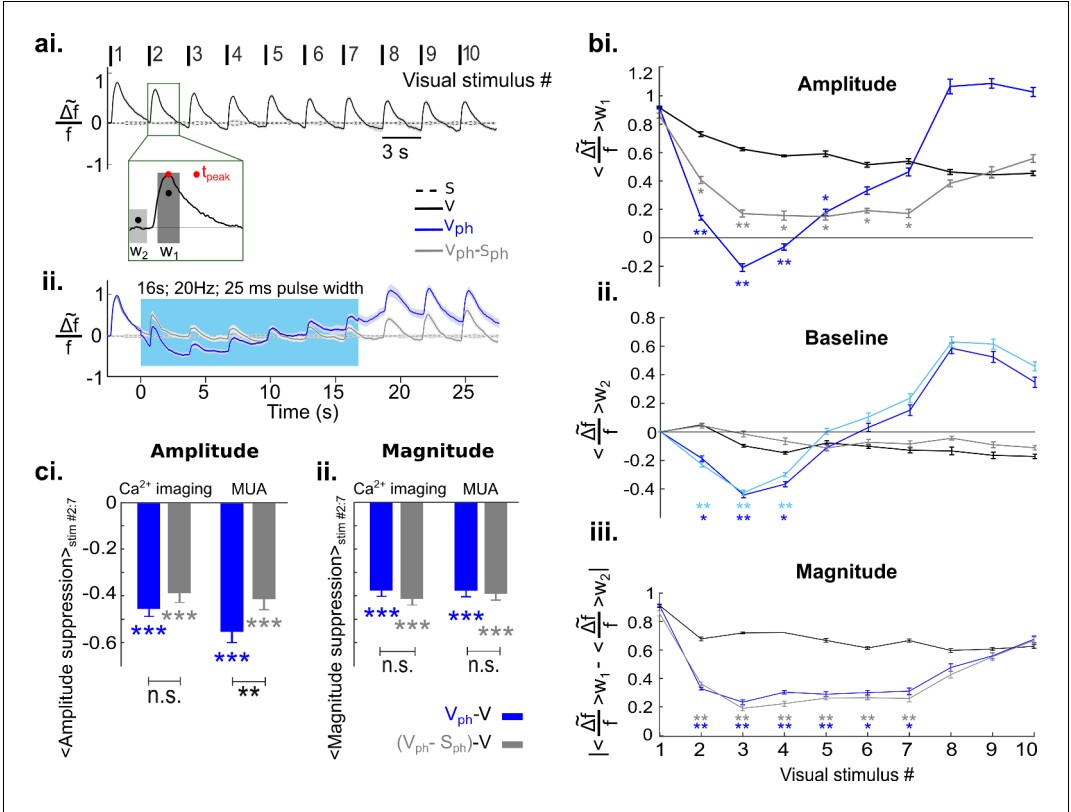

**Figure 5.** Increased serotonergic input suppresses the gain of evoked responses. (ai) Visually evoked responses in the control condition (V). Traces show spatial averages across the center of activity in V1 (see black circle in *Figure 2b*, second row) after repetitive visual stimulation (onsets and numbering at top; n=8 animals, each experiment comprises 25–50 trials). Traces are normalized to the maximum of the first visual response. Inset: $w_1$ refers to 400 ms time window around peak value and $w_2$ marks 200 ms time window before the onset of each visual stimulus. Stippled black lines show spontaneous activity without visual input and without photostimulation. (aii) Visually evoked response with concurrent DRN photostimulation ($V_{ph}$, blue trace) and evoked component (i.e. after subtraction of spontaneous activity ($S_{ph}$) from $V_{ph}$, gray trace). (bi) Amplitude values: average of activity in $w_1$ for each visual stimulus. (bii) Baseline values: average of activity over $w_2$ for each visual stimulus. Light blue curve shows the same calculation for $S_{ph}$ shown in *Figure 3ai*. (biii) Magnitude values: absolute difference between baseline values **bii** and amplitude values **bi** for each visual stimulus. (ci) Summary of amplitude differences between photostimulated conditions ($V_{ph}$ [blue] and $V_{ph}$-$S_{ph}$ [gray]) and control condition (V), see legend. Bars show the average of amplitude difference during the time of photostimulation (stimulus interval #2–7, shown in **bi**). $Ca^{2+}$ imaging: left blue and gray bars, mean ± SEM, n=8 animals. Extracellular recordings: right blue and gray bars, mean ± SEM, n=9 animals, 104 MUA. (cii) Same as **ci** for magnitude values shown in **biii**. All the traces depict mean values across animals, shaded areas (in **ai** and **aii**) as well as error bars (in **bi-biii**) represent SEM. Color legend in **ai** applies to all panels. ***p<0.001, **p<0.01, and *p<0.05, paired t test with permutation correction for multiple comparisons (**bi-biii**) and one-sample and paired t test (**ci** and **cii**). For sham control see *Figure 5—figure supplement 3*.

The online version of this article includes the following figure supplement(s) for figure 5:

**Figure supplement 1.** Extracellular recordings confirm suppressive effects observed with $Ca^{2+}$ imaging.
**Figure supplement 2.** Divisive suppression of evoked activity following photostimulation of the DRN.
**Figure supplement 3.** Photostimulation of control ePet-Cre mice injected with NaCl solution does not affect visually evoked and spontaneous activity.

averaging window also includes the rising baseline of the $Ca^{2+}$ signal (*Figure 5bii*). Consequently, the magnitude of the responses which is independent from baseline changes reveals a similar significant reduction in $Ca^{2+}$ and MUA data (*Figure 5cii*, $Ca^{2+}$ imaging data: −0.38±0.01 [blue], −0.41±0.01 [gray], n=8 animals, p<0.001, one-sample t test; MUA data: −0.38±0.02 [blue], −0.39±0.02 [gray], n=9 animals, p<0.001, one-sample t test). This again demonstrates that 5-HT-

induced reduction in the gain of evoked activity is independent of the reduction of ongoing activity and is well-captured by response magnitude in both recording methods. Analogous to spontaneous activity, using linear regression, we found that the suppression of the evoked component is divisive (*Figure 5—figure supplement 2*). Altogether, these results suggest that increasing activity of 5-HT neurons in the DRN affects cortical activity in a divisive manner via two suppressive components: one suppressing ongoing activity and another reducing the gain of visually evoked activity.

## Distinct and independent contribution of 5-HT2A and 5-HT1A receptors to suppression of evoked and spontaneous activity

We next asked whether the observed two suppressive components might be mediated via different 5-HT receptors (*Hannon and Hoyer, 2008*; *Leysen, 2004*; *Santana et al., 2004*). After blocking 5-HT2A receptors via microiontophoresis of MDL (see Methods) and parallel photostimulation of 5-HT neurons in the DRN, the amplitude of the evoked response remains suppressed, while its magnitude is only slightly reduced (*Figure 6ai*, blue). Hence, after subtraction of the spontaneous component (*Figure 6ai*, light blue), the trace of the evoked component (*Figure 6aii*, gray) is nearly identical to control conditions (*Figure 6aii*, black) with no significant difference between magnitude values

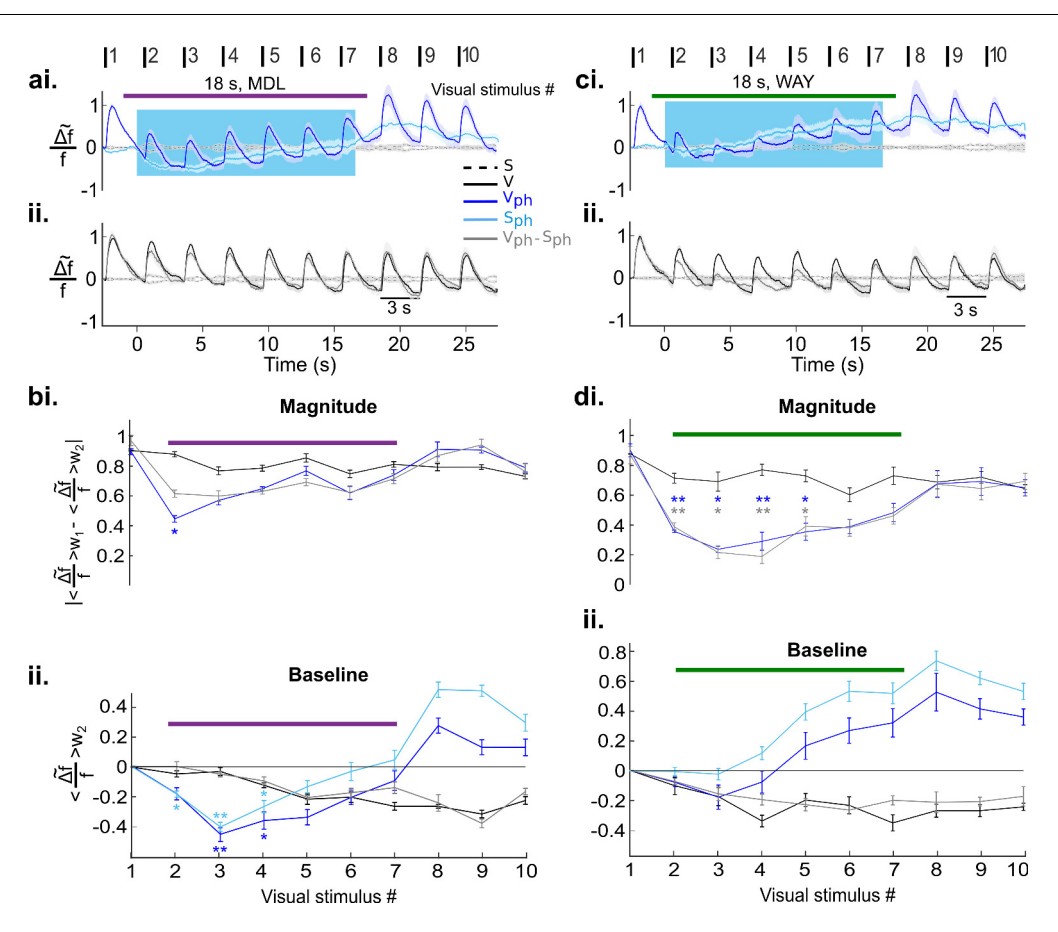

**Figure 6.** Independent and distinct suppressive effects of 5-HT2A and 5-HT1A receptors on activity in the visual cortex. (**ai**) Visual cortical response after DRN photostimulation and additional blocking of 5-HT2A receptors via microiontophoresis of MDL (magenta horizontal bar shows timing of MDL administration, 18 s; mean ± SEM, n=6 animals). $V_{ph}$ and $S_{ph}$ depicted as dark and light blue traces, respectively. (**aii**) Control visual response (V) and evoked component ($V_{ph}$-$S_{ph}$). (**bi-ii**) Quantification of the traces shown in **ai** and **aii**, values were derived as described in *Figure 5bii* (baseline) and *Figure 5biii* (magnitude). (**ci-ii**) Same as **ai** and **aii** with blocking of 5-HT1A receptors via microiontophoresis of WAY (green horizontal bar shows timing of WAY administration, 18 s; mean ± SEM, n=4 animals). (**di-ii**) Same analysis as applied in **bi** and **bii** for the traces shown in **ci** and **cii**. **p<0.01 and *p<0.05, paired t test with permutation correction for multiple comparisons.

(*Figure 6bi*) except for a period immediately after onset of drug application (*Figure 6bi*, stimulus #2). This is most likely due to delayed onset of the drug effect. In contrast, the suppression of the baseline is preserved (*Figure 6bii*, dark blue) and is similar to spontaneous activity during photostimulation and MDL treatment (*Figure 6bii*, light blue). It is noteworthy that the rising part of the RCaMP signal in the $S_{ph}$ condition during MDL application (*Figure 6bii*, light blue, values #5:7) is significantly reduced as compared to $S_{ph}$ without MDL treatment (*Figure 5bii*, light blue; p=0.04, two-sample t test on the detrended traces). This suggests a reduction of the aforementioned intracellular $Ca^{2+}$ accumulation (*Eickelbeck et al., 2019*) by effectively blocking 5-HT2A receptors. However, the continued rise of the RCaMP signal after the initial dip suggests contribution of further 5-HT receptor types (*Jang et al., 2012*; *Millan et al., 2008*) involved in intracellular $Ca^{2+}$ accumulation that remain active during MDL application (*Villalobos et al., 2005*; *Varga et al., 2009*).

Conversely, blocking of 5-HT1A receptors by microiontophoresis of WAY (see Materials and methods) abolishes the suppression in spontaneous activity (*Figure 6ci*, light blue and *Figure 6dii*), whereas the evoked component (*Figure 6cii*, gray) and hence, the magnitude of the response, remain significantly suppressed almost the entire time of drug application and photostimulation (*Figure 6di*, p<0.05, paired t test with permutation correction).

To compare 5-HT-induced effect on evoked and spontaneous components with and without drug application, time-averaged values during photostimulation are summarized in *Figure 7*. Suppression of the response magnitude (*Figure 7a*, left two bars) is significantly (p<0.01, two-sample t test) reduced via blocking 5-HT2A receptors (−0.18±0.03 [blue] and −0.16±0.03 [gray], middle two bars, n=6 animals, p>0.05; one-sample t test). No significant difference was found in magnitude of the evoked component by blocking 5-HT1A receptors (−0.36±0.05 [blue] and −0.37±0.05 [gray], right two bars; n=4 animals, p<0.001, one-sample t test). This suggests that 5-HT2A receptors dominantly contribute to the suppression of the evoked component by controlling the gain of the evoked response. In contrast, suppression of spontaneous activity (*Figure 7b*, left blue bar) is abolished via blocking of 5-HT1A receptors (0.02±0.02, n=4 animals, p>0.5, one-sample t test; *Figure 7b*, right blue bar), while no such change is seen by blocking 5-HT2A receptors (−0.4±0.02, n=6 animals, p<0.01; one-sample t test; *Figure 7b*, middle blue bar). Thus, activation of 5-HT1A receptors dominantly contributes to suppression of spontaneous activity.

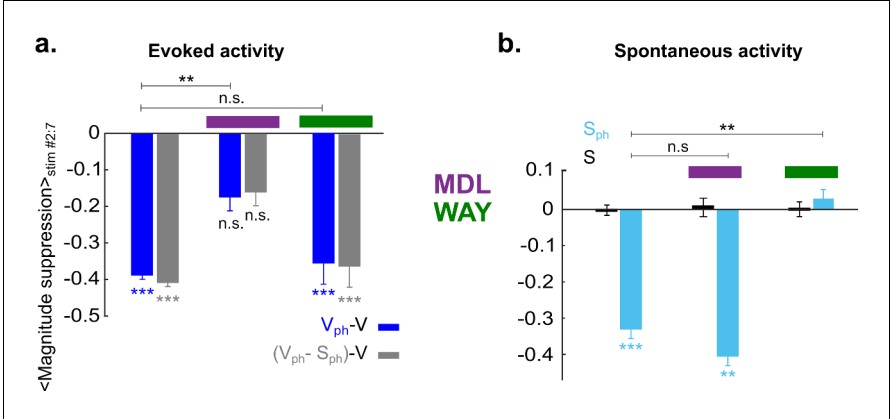

**Figure 7.** 5-HT2A and 5-HT1A receptors act independently on suppression of evoked and spontaneous components, respectively. (a) Summary of magnitude differences between photostimulated conditions ($V_{ph}$ [blue] and $V_{ph}$-$S_{ph}$ [gray]) and control condition (V). Bars show magnitude difference during the time of photostimulation (shown in *Figure 5biii*, stimulus interval #2–7). The two bars at left depict values obtained without 5-HT receptor antagonists, bars at right with use of antagonists. (b) Comparison of spontaneous activity with (blue) and without (black) photostimulation. Spontaneous activity is averaged over the time window shown by the red rectangle ($w_{red}$) in *Figure 3ai*. First two bars at left depict spontaneous activity without application of 5-HT antagonists. Experiments with application of MDL and WAY are marked with magenta and green bars, respectively. ***p<0.001 and **p<0.01, one-sample t test and two-sample t test for comparison between groups.

Together these results establish 5-HT1A and 5-HT2A receptors as major contributors to the suppression of activity in V1 induced by activation of 5-HT neurons in the DRN with separable impact on the gain of evoked and spontaneous components.

## Intensity of visual input is scaled by 5-HT-induced normalization of cortical activity

To explore how sensitively 5-HT-induced suppression affects scaling of cortical activity, we employed the well-studied sensitivity of neurons in V1 to different stimulus contrasts. To obtain a contrast response function for different stimulus intensities, we presented gratings at 100, 50, 25, 12.5, and 6.25% contrast. Responses were imaged under control conditions (*Figure 8ai*) interleaved with conditions in which the DRN was photostimulated (*Figure 8aii*). As expected from earlier studies in mice (*Porciatti et al., 1999*), visual responses progressively decline in amplitude with decreasing contrast. To obtain an objective measure of the underlying contrast tuning the maximum of evoked response (peak) for each contrast is fitted to the Naka-Rushton function (*Equation 6* in Materials and methods). The resulting fit is shown in *Figure 8di* as solid black curve ($R^2=0.93$). Upon photostimulation of the DRN, we found that overall amplitudes for each contrast are systematically suppressed (*Figure 8aii*, dark blue traces), leading to a strong downward shift of the fitted tuning curve (*Figure 8di*, solid blue, $R^2=0.93$). Furthermore, we found a systematic increase in the latency and significant decrease in the duration of evoked response (*Figure 8—figure supplement 1*). In addition, we confirmed the suppression of spontaneous activity independent of visual input (*Figure 8aii*, light blue trace). Above we showed that this suppressive spontaneous component contributes additively to the 5-HT-induced overall suppression of visual responses (*Figure 5*). Note that the evoked component is obtained by subtraction of the spontaneous component from the evoked responses for each contrast (*Figure 8aiii*). Therefore, the fitted tuning curve of the evoked component (*Figure 8di*, solid gray, $R^2=0.92$) shows a shift toward tuning of control conditions that reflects the portion of suppression in the evoked response (*Figure 8di*, solid blue) attributed to the spontaneous component.

Next, to account for normalization the traces of evoked responses (*Figure 8aii*) and evoked component (*Figure 8aiii*) are scaled by their maximum value at 100% contrast (*Figure 8ci and cii*, respectively). Such divisive scaling fails to fully replicate contrast tuning of controls (*Figure 8di*, solid black curve) when considering evoked responses, particularly at low stimulus contrasts (*Figure 8di*, blue stippled curve, $R^2=0.92$). This is because of a significant contribution of the spontaneous component to the overall suppression (i.e., gain) of evoked responses at low stimulus contrasts: Whereas the magnitude of the evoked component shows no further reduction at these lower contrasts (*Figure 8bi*, two right gray bars), the corresponding suppression in the magnitude of evoked responses (which include the suppression in spontaneous activity) are significant (*Figure 8bi*, two right blue bars; 12%: $-0.19\pm0.005$, p<0.001; 6%: $-0.15\pm0.003$, p<0.01; n=18 animals, one-sample t test). This indicates that at low contrast suppression is mainly due to the suppression in the spontaneous component. Also note that the suppression at these lower contrasts is similar to 5-HT-induced suppression of spontaneous activity ($S_{ph}$; $-0.17\pm0.02$, n=18 animals, p<0.001, one-sample t test; *Figure 8bii*). Consequently, fitting the Naka-Rushton function to the added values of evoked and spontaneous components (*Figure 8di*, gray stippled curve; $R^2=0.93$) reveals a close match to contrast tuning of control conditions (*Figure 8di*, solid black curve), indicating normalization across all contrasts. Importantly, following the same experimental paradigm and quantification methods, MUA recordings leads to equivalent findings (*Figure 8dii*, 108 MUAs, n=3 animals; see *Figure 8—figure supplement 2* for response traces and similar analysis). We conclude that in the anesthetized state, 5-HT-induced normalization of visual responses to different stimulus intensities is achieved by a linear combination of 5-HT-induced suppression of spontaneous and evoked components of cortical activity.

Finally, we explored the extent to which the above conclusion depends on the cortical state. Previous studies suggest that during awake state the cortical 5-HT levels are increased (*Mukaida et al., 2007*; *Portas et al., 2000*). Because our results characterize 5-HT-induced suppression as divisive, both for spontaneous and evoked activity (*Figure 4* and *Figure 5—figure supplement 2*, respectively), 5-HT baseline levels may crucially affect the quantity of inducible suppression. Hence, the impact of activating the DRN on cortical activity may differ under wakeful conditions compared to conditions where the animals are anesthetized. Therefore, we recorded cortical activity of awake

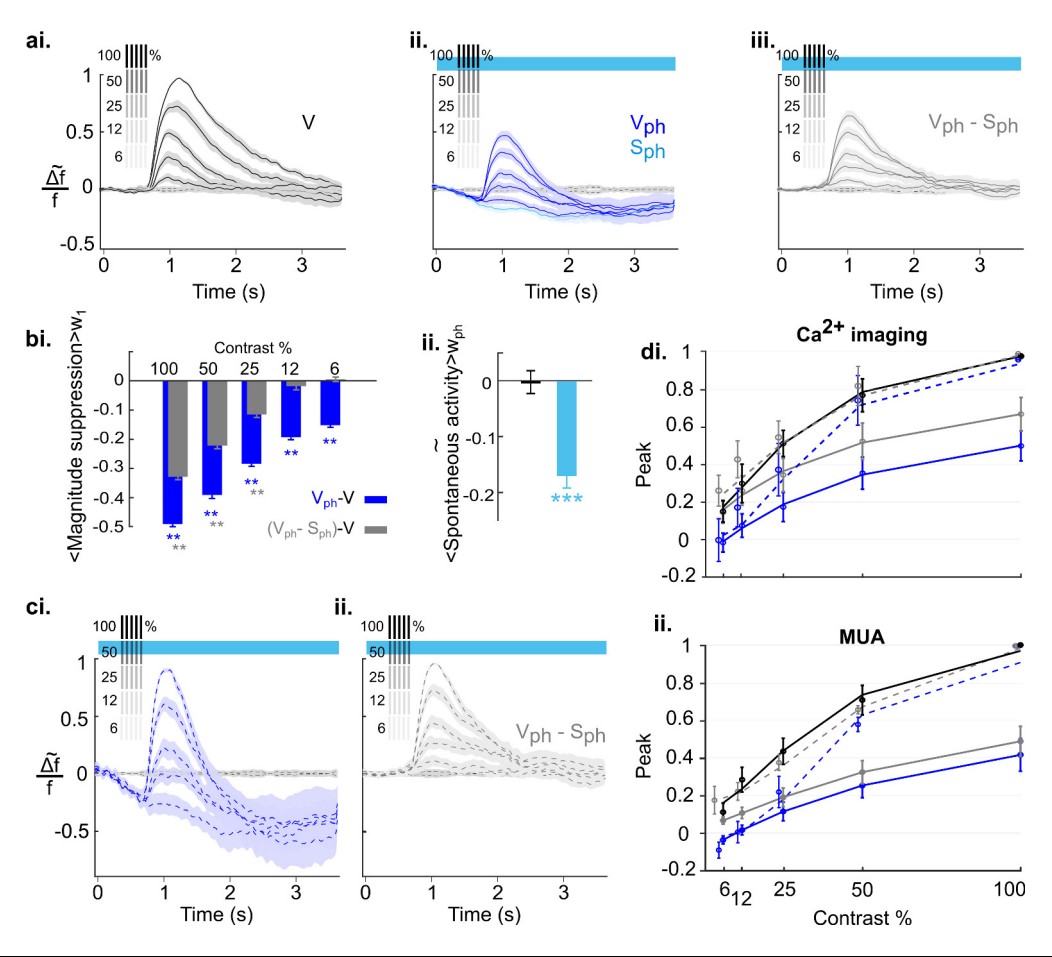

**Figure 8.** Effects of serotonergic modulation response normalization. (**ai**) Control conditions (V): visually evoked response to stimulation with gratings of different contrast. Values are scaled to the maximum amplitude at 100% contrast (n=18 animals). Grating icons identify the timing of visual stimulation and its corresponding contrast. Black dashed lines show activity devoid of photostimulation and visual input. (**aii**) Same as **ai** with concurrent DRN photostimulation. Dark blue trace depicts visually evoked response ($V_{ph}$), light blue trace is for spontaneous activity ($S_{ph}$). (**aiii**) Evoked component ($V_{ph}$-$S_{ph}$) of the traces shown in **aii**. (**bi**) Magnitude differences between photostimulated conditions ($V_{ph}$ [blue] and $V_{ph}$-$S_{ph}$ [gray]) and control condition (V) for each grating contrast. (**bii**) Average of spontaneous activity (S, black) and $S_{ph}$ (light blue) over the entire time window of photostimulation ($w_{ph}$). (**ci**) Scaling the traces in **aii** to the maximum of the trace at 100% contrast. (**cii**) Same as **ci** for the evoked component as shown in **aiii**. (**di**) Solid lines: Peak values of traces shown in **ai**-**aiii** fitted to the Naka-Rushton function (see Methods). Stippled lines: Fit of the peak values of the normalized traces shown in **ci** and **cii** to the Naka-Rushton function. (**dii**) Contrast tuning obtained from extracellular recordings (3 animals, 108 MUA). Similar to **di**, the peak values of traces for each contrast and condition are fitted to Naka-Rushton function (all traces are shown in *Figure 8—figure supplement 2ai—aiii, ci and cii*). \*\*p<0.01 and \*\*\*p<0.001, one-sample t test. The online version of this article includes the following figure supplement(s) for figure 8:

**Figure supplement 1.** Influence of serotonergic input on latency and duration of evoked responses.
**Figure supplement 2.** MUA revealed the same signatures of 5-HT-induced response normalization as RCaMP imaging.

mice (5) that were head-fixed and habituated to walk and stay on a treadmill (using the same experimental paradigm as shown in *Figure 8*). *Figure 9a* depicts the time traces of spatially averaged RCaMP signals across V1, evoked by different stimulus contrasts. As in anesthetized animals, following DRN photostimulation cortical activity was suppressed as compared to controls (*Figure 9ai and*

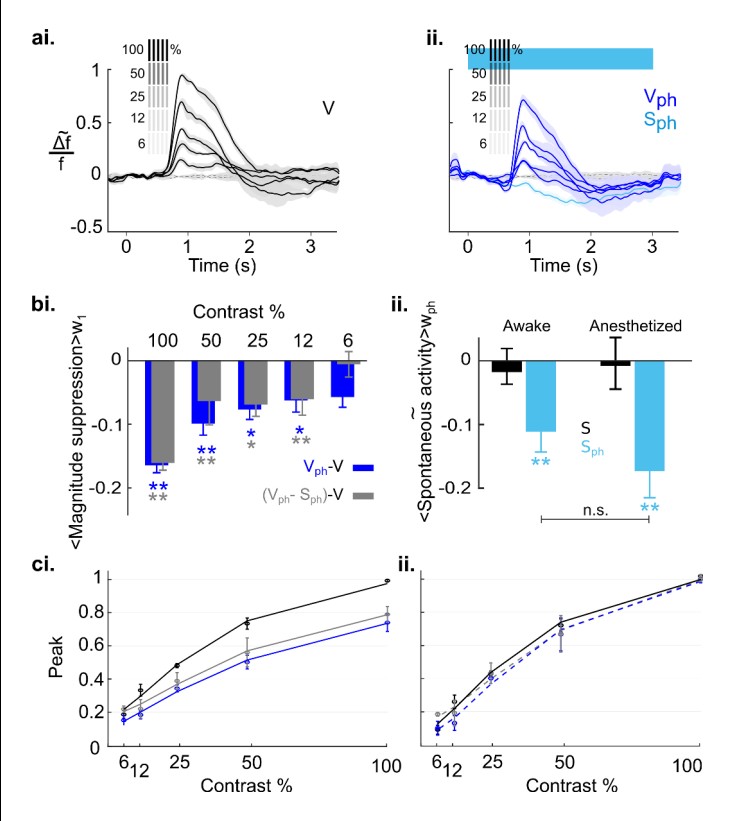

**Figure 9.** Serotonergic normalization effects in the awake state. Same conventions as in *Figure 8*: (a) Visually evoked responses (ai) Control conditions (V). (aii) same as ai with concurrent photostimulation of the DRN (V_ph, dark blue). Light blue trace shows spontaneous activity during photostimulation (S_ph). (bi) Magnitude differences between photostimulated conditions (V_ph [blue] and V_ph-S_ph [gray]) and control condition (V) for each grating contrast. (bii) Average of spontaneous activity (S, black) and spontaneous activity with photostimulation (S_ph, light blue) in awake and anesthetized state. Values are averaged across the time window of photostimulation (w_ph). **p<0.01, and *p<0.05, one-sample t test. (ci) Peak values of traces shown in ai and aii fitted to the Naka-Rushton function. Line colors correspond to colors of traces in ai and aii, gray depicts peak values and fit of the evoked component (traces not shown). (cii) Same as ci with normalized peak values. All data summarize recordings in five awake mice. Traces of matched animals in the anesthetized state are shown in *Figure 9—figure supplement 1*. For differences in the contribution of evoked and baseline components in normalization see *Figure 9—figure supplement 2*.

The online version of this article includes the following figure supplement(s) for figure 9:

**Figure supplement 1.** Recordings of the same mice shown in *Figure 9*, here under anesthesia (n=5 animals).

**Figure supplement 2.** Weights of evoked and baseline components in normalization depend on cortical state.

*aii*, blue and black (control) traces, respectively). In comparison to the anesthetized state, however, suppression at higher contrast (100%–25%) is significantly reduced (100%: p=0.010, 50%: p=0.012, 25%: p=0.019, 12%: p=0.392, 6%: p=0.694, n=5 animals, two-paired t test; *Figure 9—figure supplement 1* shows the data derived from the same animals when they were anesthetized). Importantly, in the awake state, the contrast-dependent decline in the magnitudes is not different between suppression of evoked activity and evoked component (*Figure 9bi*, blue and gray bars, respectively; p>0.31, paired t test). This suggests that normalization is largely controlled without the contribution of the spontaneous component (*Figure 9ci*, blue ($R^2$=0.98) and gray ($R^2$=0.95) curves for evoked response and evoked component, respectively). Indeed, fitting normalized data (i.e. traces scaled by their maximum value at 100% contrast) to the Naka-Rushton function reveals a similar contrast tuning function for evoked responses and evoked component (*Figure 9cii*, stippled blue

($R^2$=0.97) and gray ($R^2$=0.98) curves, respectively) as controls (*Figure 9cii*, solid black curve). Thus, in the awake state, suppression of spontaneous activity ($-0.11\pm0.3$, n=5 animals, p=0.022, one-sample t test; *Figure 9bii*, left blue bar), even though similarly significant as in the anesthetized state ($-0.17\pm0.04$, n=5 animals, p=0.005, one-sample t test; *Figure 9bii*, right blue bar), constitutes an independent component with little influence on the magnitude and consequently, on normalization of stimulus-evoked responses.

## The weight of spontaneous activity in response normalization differs between awake and anesthetized states

To understand the different relative impact of spontaneous activity on response normalization in the two cortical states, the amount of 5-HT-induced suppressive gain should be considered as a crucial adjusting factor. This factor influences not only the weight of the evoked component but also the weight of the spontaneous component. Assuming that the control visual response (V) can be approximated by the visual response during photostimulation ($V_{ph}$) and the 5-HT-induced gain (i.e. the inverse of c), it can be formalized:

$$\frac{V_{ph}}{V} = c \rightarrow V = \frac{1}{c} \times V_{ph} \tag{1}$$

Our main hypothesis here is that $V_{ph}$ is a linear combination of the evoked component ($E_{ph}$) and its baseline component ($b_{ph}$). As shown in *Figure 5bii*, $b_{ph}$ is approximately equal to the suppression in spontaneous activity ($S_{ph}$). Furthermore, we show (*Figure 5bi*) that the general effect of photostimulation is suppressive (i.e., g>1):

$$\begin{cases} V_{ph} = E_{ph} + b_{ph} \\ b_{ph} \cong S_{ph} \\ \frac{1}{c} = g \overset{c<1}{\rightarrow} g>1 \end{cases} \tag{2}$$

Thus, under these considerations *Equation 1* can be rewritten as:

$$V = g(E_{ph} + S_{ph}) \tag{3}$$

$$V = g \times E_{ph} + g \times S_{ph} \tag{4}$$

formalizing that the gain (g) affects the relative contributions of both the evoked and the spontaneous component.

Next, we focus on how normalization affects the relative weight of the baseline component ($g \times S_{ph}$) dependent on the cortical state (*Figure 9—figure supplement 2ai and aii*). In the anesthetized state, the weight of the baseline component is significantly larger (-0.36±0.11, n=18) than in the awake condition (-0.11±0.02, n=5; p<0.001, two-sample t test). This influences normalization particularly at low contrasts, where the weights of the baseline and the evoked components are not significantly different from each other (*Figure 9—figure supplement 2bi*). Thus, in the anesthetized state, the baseline component is a likely candidate for an additive contribution to normalization (*Figure 8di*). In contrast, in the awake state, the linear summation of both components (*Equation 4*) is significantly (p<0.01, one-sample t test) biased towards the evoked component, with negligible contributions of the baseline component across all stimulus intensities (*Figure 9—figure supplement 2bii*).

One might argue that the reduced contribution of the baseline component in the awake state is merely due to the smaller value of baseline suppression (i.e. $S_{ph\ aw} < S_{ph\ an}$) at the time of the peak ($t_{pk}$) of the evoked response. In fact, we find that $V_{ph}$ in the awake state reaches peak intensities significantly earlier than in the anesthetized state (mean over all contrasts: 311±32 ms (n=5 awake mice), 441±41 ms (n=18 anesthetized mice), p<0.001 two-sample t test; *Figure 9—figure supplement 2ai*). Note, however, that the amount of suppression in $S_{ph}$ at $t_{pk}$ in the awake state, is still significant (-0.09±0.03, n=5, p<0.01, one-sample t test), although less than in the anesthetized state (-0.17±0.03, n=18, p<0.001, two-sample t test). In any case, as shown in *Figure 9bii*, the overall amount of suppression of $S_{ph}$ in both states is not significantly different when averaged over the

entire time of photostimulation. Therefore, first, it follows that suppression in spontaneous activity in the awake state is strictly either lower or equal than in the anesthetized state (*Equation 5*, first line).

$$\begin{cases} S_{phaw} \leq S_{phan} \\ g_{aw} < g_{an} \end{cases} \tag{5}$$

Second, we find that suppression of the magnitude is constant during photostimulation (*Figure 5biii*) and always lower in the awake (0.78±0.1, n=5) as compared to the anesthetized state (0.52±0.2, n=18, p<0.001, two-sample t test; *Figure 8bi* and *Figure 9bi*). Together, it is conclusive that the 5-HT-induced gain is smaller in the awake than in the anesthetized state ($g_{aw} < g_{an}$, *Equation 5*, second line).

Ultimately, *Equation 5* implies that the weight of the baseline component is generally lower in the awake than in the anesthetized state [$(g \times S_{ph})_{aw} < (g \times S_{ph})_{an}$]. In *Figure 9—figure supplement 2biii.* a condition with equal suppression of spontaneous activity in both states ($S_{ph\ aw} = S_{ph\ an}$) is tested. Still, the baseline component in the awake state does not significantly contribute to normalization. In addition, as stated above, given that in the awake state the evoked component ($g \times E_{ph}$) is significantly higher than the spontaneous component ($g \times S_{ph}$) across all contrasts (*Figure 9—figure supplement 2bii*), *Equation 4* can be written as $V \cong g \times E_{ph}$. This states that in the awake state normalization is largely controlled by the gain of the evoked component. Altogether, this supports our conclusion that in the awake state, through a reduction in gain, the relative contribution of components to normalization is biased towards the evoked component and largely achieved without the requirement of a significant additive contribution of the baseline component, as is the case in the anesthetized state (*Figure 8di, Figure 9ci and cii*).

In summary, in contrast to the anesthetized state, under awake conditions normalization of cortical responses to different visual contrasts is largely conveyed by the gain of the evoked component. Still, we find a concurrent suppression of spontaneous activity constituting a separable component, which has, in contrast to the anesthetized state, only minor additive effects on normalization of responses across varying stimulus intensities.

## Discussion

We show that activation of the serotonergic system impacts two entities of population activity in the visual cortex (V1): internally ongoing (spontaneous) and visually driven (evoked) responses, affecting both in a separable and divisive manner. Each of these components is scaled through separate suppressive effects of 5-HT1A and 5-HT2A receptors, respectively.

Previous studies in V1, using specific agonists for 5-HT1B and 5-HT2A receptors combined with single-unit recordings in anesthetized monkey, found a bi-directional modulation dependent on instantaneous firing levels. Application of 5-HT2A agonist caused suppression of neurons with strong responses and facilitation of those with weak responses, the opposite occurred when applying 5-HT1B receptor agonist (*Shimegi et al., 2016*; *Watakabe et al., 2009*). At the population level, 5-HT microiontophoresis in awake monkey revealed predominant suppressive effects (*Seillier et al., 2017*), similar to our study. In contrast to our findings, suppression was restricted to divisive scaling of evoked responses while no change in spontaneous activity was observed (*Seillier et al., 2017*). This discrepancy may indeed point to species-specific differences in individual 5-HT receptor sets or 'receptomes'. On the other hand, limitations associated with local iontophoretic 5-HT application, such as determining the exact onset of drug effects or the inevitable lack of cortical stimulation with layer- and cell type-specific synaptic weights of DRN synapses could be an alternative explanation. Here, we applied optogenetic approaches to activate 5-HT neurons in the DRN and to specifically target the serotonergic system as a whole circuit (*Jacobs and Azmitia, 1992*; *Ogawa et al., 2014*; *Pollak Dorocic et al., 2014*; *Weissbourd et al., 2014*) in order to explore subsequent changes in V1. Considering the broad topographic organization within the DRN (*Muzerelle et al., 2016*), other cortical and subcortical areas modulated by activation of 5-HT neurons may additionally influence activity in V1 (*Gilbert and Li, 2013*). It should also be noted that besides 5-HT neurons, subpopulations of glutamatergic cells in the raphe nuclei express ePet (*McDevitt et al., 2014*; *Sos et al., 2017*). ePet in non-serotonergic neurons is mainly observed in the median raphe nucleus (*Pelosi et al., 2014*; *Sos et al., 2017*), while in the dorsal and caudal nuclei (B6 and B7) only few

neurons positive for ePet are non-serotonergic (*Pelosi et al., 2014*). Thus, by using the ePet-Cre mouse line and the fact that 5-HT-neurons may use glutamate co-transmission (*Kapoor et al., 2016*; *Liu et al., 2014*; *Szőnyi et al., 2016*), an indirect activation of other (none-5-HT) circuits could principally be involved in the observed suppressive effects. However, as seen in our experiments pharmacological blocking of 5-HT receptors in V1 largely diminished the induced suppressive effects, contribution from other modulatory systems appears negligible for the present study.

The DRN was photostimulated for 16 s in the first set of our experiments. This is relatively long in comparison to the brisk activation of DRN neurons observed in certain behavioral contexts (*Heym et al., 1982*; *Ranade and Mainen, 2009*). However, this does not exclude additional longer-lasting tonic increase in DRN activity under natural conditions. For instance, direct retinal projections to the DRN (*Pickard et al., 2015*; *Ren et al., 2013*; *Zhang et al., 2016*) suggest its regulation well beyond the stimulation time window chosen in our study. In addition, slow periodic changes in 5-HT release occur naturally during day-night cycle (*Portas et al., 2000*; *Tyree and de Lecea, 2017*) and depend on seasonal factors (*Spindelegger et al., 2012*). Importantly, using the ePet-Cre mouse model and comparable photostimulation parameters as in our experiments (i.e. similar blue light intensity and frequency) to activate the DRN via fiber optics, a consistent increase of DRN activity in fMRI and MUA recordings over 20 s of photostimulation was revealed (*Grandjean et al., 2019*). Prolonged (12 s) photostimulation of the DRN in SERT-Cre mice increased perceptual thresholds for tactile stimuli (*Dugué et al., 2014*), altogether suggesting that sustained or repeated elevation of DRN activity correlates with meaningful and behaviorally relevant changes in neuronal responses (*Correia et al., 2017*; *Lőrincz and Adamantidis, 2017*; *Miyazaki et al., 2011*).

We used wide-field RCaMP imaging, an optogenetic method that enables recordings of suprathreshold neuronal activity across several millimeters of cortical target areas (*Kirmse et al., 2015*; *Michaiel et al., 2019*; *Shimaoka et al., 2019*; *Silasi et al., 2016*; *Xiao et al., 2017*), carrying information encoded by large populations of neurons (*Dayan and Abbott, 2001*). Wide-field recordings of RCaMP signals may also correlate with activity in the neuropil (*Allen et al., 2017*). To rule out this possibility, we performed complementary recordings of MUA. These largely reproduced the imaged suppressive effects, indicating that the observed reduction of RCaMP signals indeed reflects reduced spiking output at the cortical population level. Our recorded RCaMP signals also showed an increase above the baseline level before the end of DRN photostimulation. Autoreceptor-mediated suppression of 5-HT neurons within the DRN is an unlikely explanation for producing such late increase. First, the applied photostimulation parameters induce sustained elevation in activity of 5-HT neurons in the DRN during the entire time window of photostimulation (*Figure 1—figure supplement 1* and see *Grandjean et al., 2019*). Second, we show using MUA that spiking activity in V1 is consistently suppressed during photostimulation. Third, the evoked component of visual responses remains suppressed during DRN photostimulation and finally, the late increase of RCaMP signals was selectively reduced by 5-HT2A antagonist. We suggest that intracellular $Ca^{2+}$ accumulation of cortical neurons produces the late rise in the RCaMP signal (*Eickelbeck et al., 2019*) due to 5-HT receptor-mediated activation of the Gq/11 pathway (*Jang et al., 2012*; *Millan et al., 2008*) with further activation of store-operated channels (*Celada et al., 2013*). Additionally, $Ca^{2+}$ signals could increase through activation of 5-HT3 receptors (*Nichols and Mollard, 1996*) that are also capable of providing fast disynaptic inhibition via activation of interneurons (*Varga et al., 2009*). Finally, the late increase of the RCaMP signal did not affect our conclusions but may be considered when using wide-field $Ca^{2+}$ recordings of cortical activity in future studies.

Using different stimulus intensities, we identified normalization of visual responses. In anesthetized animals, ongoing activity supplemented response normalization as a subtractive factor. In the awake state, normalization was controlled by the gain of the evoked component and independent of concomitant suppression in ongoing activity. Inhibition tends to dominate activity in awake cortex (*Haider et al., 2013*), possibly accompanied by increased 5-HT levels (*Mukaida et al., 2007*; *Portas et al., 2000*). Therefore, because suppression of spontaneous activity is divisive (i.e., dependent on initial baseline levels), the additional contribution of the spontaneous component to normalization in the anesthetized state may result from large fluctuations in its amplitude rarely exhibited during wakefulness (*Haider et al., 2013*). Thus, the degree of 5-HT-controlled integration of sensory input and ongoing activity depends on cortical state and initial 5-HT levels.

Our pharmacological blocking experiments suggest that (hyperpolarizing) 5-HT1A receptors convey the suppression of the spontaneous component to a large extent. However, both RCaMP

imaging and measurements of multi-unit activity, as applied here, do not allow identifying cell type specific contributions. At the population level, suppression of ongoing activity may reflect an overall reduction of pyramidal cell spiking caused by hyperpolarization or by decrease in synaptic transmission (*Lőrincz and Adamantidis, 2017*). In fact, hyperpolarization of excitatory neurons was recently shown as an additional source of normalization processes (*Sato et al., 2014*). In contrast, divisive scaling of evoked responses is most likely conveyed by predominant activation of (excitatory) 5-HT2A receptors, as suppression of the evoked component largely diminished after specific blocking. Moreover, 5-HT2A receptors couple to the Gq/11 pathway (*Hannon and Hoyer, 2008*) leading to an increase in neuronal firing rather than suppression. Therefore, activation of GABAergic neurons may mediate reduction in activity of pyramidal neurons (*Gellman and Aghajanian, 1993*). In fact, divisive modulation of visual cortical responses was shown to be specifically dependent on activation of soma-targeting parvalbumin expressing interneurons (*Wilson et al., 2012*, but see *Lee et al. (2012)* and *Seybold et al., 2015*) that dominantly express 5-HT2A receptors (*Puig and Gulledge, 2011*; *Weber and Andrade, 2010*). Activation of 5-HT2A receptors may also cause concomitant increase in depolarizing currents in pyramidal neurons, producing shunting inhibition, which results in increased conductance. Such a mechanism is known to affect the gain and the time constant of neuronal responses (*Carandini and Heeger, 2012*). Consistent with shunting inhibition, we find that response-onset latency decreases as a function of stimulus contrast while response duration in both control and photostimulated conditions increases. Furthermore, we find the duration of responses declined during photostimulation of the DRN as compared to controls. This suggests a shortened time window for temporal summation that in turn reduces response amplitudes (*Mante et al., 2008*). Thus, reduction in response duration under the influence of 5-HT may indicate involvement of shunting inhibition in pyramidal neurons to effectively control response gain. Suprathreshold excitatory drive by 5-HT2A activation may further be amplified by shaping the duration of 5-HT1A-mediated inhibition (*Avesar and Gulledge, 2012*; *Stephens et al., 2014*). Altogether, we propose that 5-HT circuits modulate evoked visual population responses by partly plugging into the existing canonical machinery performing divisive normalization (*Atallah et al., 2012*; *Carandini and Heeger, 2012*; *Lee et al., 2012*; *Wilson et al., 2012*). Further studies are needed to clarify 5-HT-dependent microcircuit mechanisms at the single-cell level (*Tang and Trussell, 2017*).

In two recent studies, selective activation of 5-HT2A receptors produced a strikingly consistent suppressive effect on the gain of visually evoked population responses (*Eickelbeck et al., 2019*; *Michaiel et al., 2019*), despite cell-type and layer-specific differences across single cells (*Michaiel et al., 2019*). These results strongly support our current finding of DRN-triggered scaling of evoked responses by cortical 5-HT2A receptors at the neuronal population level. It also implicates that the distribution of a single neurotransmitter receptor density ('receptome') can account for a distinct function in sensory processing (*Akimova et al., 2009*; *Carhart-Harris et al., 2016*; *Deco et al., 2018*; *Goldberg and Finnerty, 1979*; *González-Maeso et al., 2007*; *Tauscher et al., 2001*). While 5-HT2A receptors control response gain, DRN-triggered scaling of ongoing V1 activity seems dominantly controlled by 5-HT1A receptors.

In summary, joint action of 5-HT1A and 5-HT2A receptors unfold a separable and powerful scaling of ongoing and evoked components of population activity in V1. A major difference that we observe in normalization between awake and anesthetized states is an overall stronger 5-HT-induced suppression of response gain in the anesthetized state, which implicates that in the awake state response normalization is less dependent on ongoing activity and is possibly less influenced by internal cortical broadcasts. Considering that ongoing activity contains (top-down) internal expectations, whereas evoked responses carry (bottom-up) signals about external sensory events, imbalance in the recruitment of these receptors (e.g. through specific agonist intake or disordered receptor expression pattern) affects integration of these components, and thus, cortical information flow. This may either lead to overemphasis of internally generated expectations (i.e. favoring 'priors' *Berkes et al., 2011*; *Fiser et al., 2010*) relative to sensory input or vice versa (*Lottem et al., 2016*). Long-term malfunction of such interplay facilitates psychiatric disorders like anxiety, depression and schizophrenia (*Friston, 2005*; *Geyer and Vollenweider, 2008*; *Jardri et al., 2016*; *Lucki, 1998*; *Soubrié, 1986*; *Urban et al., 2016*; *Zhang et al., 2016*). Our study may help to develop new ways of diagnosis and therapy (*Carhart-Harris et al., 2018*) by rebalancing of these components.

# Materials and methods

## Key resources table

| Reagent type (species) or resource | Designation | Source or reference | Identifiers | Additional information |
|---|---|---|---|---|
| Genetic reagent (*Mus musculus*) | ePet-Cre (B6.Cg-Tg(Fev-cre)1Esd/J) | *Scott et al., 2005* Jackson laboratory | JAX: 012712 | Breeded by the group of Dr. Stefan Herlitze and Dr. Melanie D. Mark (Ruhr University Bochum) |
| Recombinant DNA reagent | AAV1.EF1.dflox.hChR2 (H134R)-mCherry. WPRE.hGH | Addgene | Addgene number: 20297-AAV1 | AAV virus constructed to express in the Cre-expressing cells. |
| Recombinant DNA reagent | AAV1.syn.jRCaMP1a. WPRE.SV40 | Addgene *Akerboom et al., 2013* | Addgene number: 10848-AAV1 | AAV virus constructed to express in both excitatory and inhibitory neurons, due to the Synapsin promoter. |
| Antibody | Mouse anti-TPH (monoclonal) | Sigma-Aldrich | T0678-100ul | 1:200 |
| Antibody | Anti-mouse DyLight 488 (donkey polyclonal) | Abcam | | 1:500 |
| Antibody | Anti-c fos (rabbit polyclonal) | Santa Cruz Biotechnology | Sc-52 | 1:1000 |
| Antibody | Anti-rabbit Alexa 488 (donkey polyclonal) | Life Technology | | 1:500 |
| Chemical compound, drug | 5-HT2A receptor antagonist MDL-100907 | Sigma-Aldrich | M3324-5MG | 5-HT2A receptor antagonist, 20 mM in 0.9% NaCl, pH 10 |
| Chemical compound, drug | 5-HT1A receptor antagonist WAY-100135 | Sigma-Aldrich | W1895-5MG | 5-HT1A receptor antagonist, 5 mM in 0.9% NaCl pH 4 |
| Software, algorithm | MATLAB | Mathworks | RRID:SCR_001622 | |
| Software, algorithm | ImageJ | ImageJ (http://imagej.nih.gov/ij/) | RRID:SCR_003070 | |
| Software, algorithm | Inkscape | Inkscape (http://inkscape.org/) | RRID:SCR_014479 | |
| Other | Imager 3001 | Optical Imaging Inc, Mountainside, NY, USA | | |

## Mice

Adult ePet-Cre mice [ePet-Cre is a transgene with Cre recombinase driven by a serotonergic specific *ePet-1* enhancer region (*Scott et al., 2005*) were used in this study. After preparatory surgery (as detailed below), mice were housed individually and kept in 12 h light/dark cycle with food and water ad libitum.

## Viral injections and implant of optical fiber

Cre-dependent AAV [AAV1.EF1.dflox.hChR2(H134R)-mCherry.WPRE.hGH, Addgene number: 20297] was injected into the DRN of ePet-Cre transgenic mice (*Figure 1ai*) and a viral construct of the red-shifted calcium indicator RCaMP [AAV1.syn.jRCaMP1a.WPRE.SV40 Addgene number: 10848] (*Akerboom et al., 2013*) was injected into visual and somatosensory cortex based on stereo-tactic coordinates (*Lowery and Majewska, 2010*; *Paxinos and Franklin, 2004*). All viral constructs were obtained from the University of Pennsylvania (100 µL at titer $\geq 1 \times 10^{13}$ vg/mL).

Animals were anesthetized with isoflurane (4% induction and 2% for maintenance) via a nose mask and received a 0.25 ml subcutaneous bolus of isotonic 0.9% NaCl solution mixed with Bupre-norphine (10 µg/ml) and Atropine (5 µg/ml). A heating pad (37°C) was placed below the animal dur-ing surgery and experiments to maintain body temperature. Before sagittal incision along the

midline, 2% Lidocaine was applied to provide additional local anesthesia. The skull was thinned until surface blood vessels became visible. Next, a small craniotomy was made, −0.5 mm Anteroposterior [AP] to Lambda and 0 mediolateral [ML] to Bregma. Using a micromanipulator, a customized glass pipette attached to a 20 ml syringe was lowered into the brain to a depth of 2.5 mm below the brain surface to target the DRN. The viral solution containing ChR2 construct was delivered via small pressure injections (100 µm steps upwards until the depth of 1.7 mm, with an injection interval of 5 min). After injections, a custom-made optical fiber (200 µm, 0.37 NA, Thorlabs) attached to a ceramic ferrule (Thorlabs) was implanted in the brain tissue in depth of −1.5 mm below the cortical surface and 0.5 mm anteroposterior (AP) to Lambda. After implantation, the ferrule was secured to the skull with transparent dental cement (Super Bond C and B set, Hentschel-Dental, Germany). The viral solution containing RCaMP construct was injected at two locations in the visual cortex (−4.2 AP, 2.5 ML, and −3.5 AP, 2 ML to Bregma; all values in mm) and at one location in somatosensory cortex (−1.5 AP, 2.5 ML to Bregma; values in mm). At each of these cortical injection sites, 0.5 µl of the viral solution was delivered at the cortical depth of ~600 µm and 300 µm in four steps with 10 min intervals between each injection step. The thinned and exposed skull was covered with transparent dental cement and nail polish. Finally, a head holder was attached (Pi-Ku-Plast HP 36, Breedent) to the skull in order to provide a clear and easily accessible imaging window for chronic experiments. Control animals matching genetic background and age received injections with 0.9% NaCl solutions.

## Visual stimuli and DRN photostimulation

For photostimulation of serotonergic neurons in the DRN, pulses of blue light (20 Hz, 25 ms pulse width, 470 nm, using a LED driver and emitter system, Plexon) was delivered via an optical fiber attached to the implant. Photostimulation started 2.5 s after the onset of data recording and lasted for 16 s. The power of light at the tip of the fiber was ~1 mW. The used photostimulation frequency and duration have been shown to evoke a robust and sustained increase of activity in the DRN above the baseline (*Dugué et al., 2014*; *Grandjean et al., 2019*).

Vertical square-wave gratings (0.04 cycle/deg) moving at 2 Hz were presented on a monitor (100 Hz, mean luminance 40 cd/m$^2$, Sony Triniton GDM-FW900, Japan) with its center placed 30 cm away from the eye that was contralateral to the cortical recording site, overall covering ~40×60 deg of the visual field. Eyes were covered with semipermeable zero power contact lenses to prevent them from drying out or developing corneal edema. Each experiment comprised 25–50 trials (i.e., repetitions of stimulus conditions). Each trial consisted of four different conditions presented in pseudorandom order: (1) Blank condition (S), during which a uniform isoluminant gray screen was shown. These recordings were repeated twice within a trial and served as a measure of spontaneous activity. Blanks were also used to calculate relative changes in fluorescence (Δf/f, see Data analysis). (2) Visually evoked condition (control, V), during which moving gratings were repeatedly (10 times) presented with a duration of 200 ms at an interval of 3 s (during which the blank was presented). (3) Visually evoked condition during which the DRN was photostimulated ($V_{ph}$). (4) Spontaneous condition during which the DRN was photostimulated ($S_{ph}$). Each condition lasted 30 s (including 200 ms pre stimulus time), and the time interval between conditions was 60 s.

In the set of experiments where different visual contrasts were used (100, 50, 25, 12.5, 6.25%), a grating stimulus was presented once for 200 ms. In conditions comprising photostimulation, its onset was at 700 ms before the onset of the visual stimulus and lasted until the end of the trial. Additionally, spontaneous activity with and without photostimulation was recorded with identical timing to conditions comprising a grating stimulus. All conditions were presented in pseudorandomized order (interstimulus interval was 60 s).

During all recordings in the anesthetized state, mice were kept under mild anesthesia (0.5–1% isoflurane) delivered via a nose mask.

## Imaging of fluorescent RCaMP signals

Image frames (one pixel covered ~67 µm of the cortical surface followed by an additional 3×3 binning online) were collected at a rate of 100 Hz using an Imager 3001 system (Optical Imaging Inc, Mountainside, NY). The camera was focused on ~300 µm below the cortical surface. To record changes in fluorescence of the RCaMP indicator, the brain was illuminated with the excitation wavelength (560±20 nm), and the emission light (>585 nm) was collected via a dichroic mirror followed by

a bandpass filter (630±37 nm). Pre-processing and further data analysis were performed offline using custom-written scripts in MATLAB.

## Pharmacology

Microiontophoresis of MDL-100907 (Sigma-Aldrich; 5-HT2A antagonist, 20 mM in 0.9% NaCl, pH 10) or WAY-100135 (Sigma-Aldrich; 5-HT1A antagonist, 5 mM in 0.9% NaCl pH 4) was performed via a pipette inserted to the brain tissue in the region of interest (ROI). Although WAY has been found to act as an agonist at 5-HT1B and 5-HT1D receptors as well (*Davidson et al., 1997*), it is currently the most specific agent available to block 5-HT1A receptors (*Fletcher et al., 1995*; *Forster et al., 1995*). Before microiontophoresis, a ROI was selected manually as the region within V1 that showed the highest activity levels in response to a short (200 ms) visual stimulus recorded over 3 s. A small craniotomy was then made within the ROI, and the pipette, filled with MDL or WAY solutions, was inserted 300 μm below the surface of the brain. Microiontophoresis was performed concurrently with RCaMP imaging and DRN photostimulation keeping all conditions unaltered. The drug solution was retained in the pipette by applying a −10 nA retention current using a constant-current pump (Union-40 microiontophoresis pump; Kation Scientific). To apply the drugs, ejection current was delivered at 60 nA one second before the onset of DRN photostimulation and delivered continuously for 18 s. Like all other conditions, the interval between conditions was 60 s. We noticed that in each trial, the response to the first visual stimulus of the stimulus train was identical (also to controls), indicating that the inter-trial interval was of enough time to allow the abolishment of antagonists between each trial. Control experiments were done by microiontophoresis of 0.9% of NaCl solution with a pH of 4.

## In vivo extracellular multi-unit recording

A fraction of the extracellular multi-unit activity (MUA) was recorded using tungsten electrodes (0.127 mm diameter, 1 MΩ, WPI, FL, USA), at cortical depths between 250 and 750 μm. Neuronal activity was amplified 1,000× and band-pass filtered (0.2–5 kHz, Thomas Recording, Germany). The signal was recorded at a sampling rate of 20 kHz using a CED Micro1401 controlled by the Spike two software (Cambridge Electronics Design, Cambridge, UK). Spike detection was performed with the Spike two software, using a threshold above and (or) below the baseline. Additional MUA was recorded using acute 16-channel silicon probes (ASSY-1 E-1, Cambridge NeuroTech, Cambridge, UK). Signals were amplified with a 16-channel amplifier board (RHD2132, Intan Technologies, CA) and recorded using a multichannel electrophysiology acquisition board (Open Ephys, *Siegle et al., 2017*) filtered between 600 to 6 kHz. Isolated spikes were detected and clustered using the Klusta suite (*Rossant et al., 2016*; https://github.com/klusta-team/klustakwik) with default configuration parameters.

Traces of spike counts were averaged over time (200 ms bins) and across trials for each recorded condition. To normalize spontaneous activity, the individual traces of spontaneous activity with and without photostimulation were divided by the average spike counts over 1 s after the onset of recording. In order to normalize the evoked responses, an average spike count over 1 s after the onset of recordings was subtracted from each trace. For further normalization, the traces for each condition were divided by the response amplitude to the first stimulus in control (visually evoked, V) condition. The normalized traces were then averaged over all the units and animals. The evoked component of $V_{ph}$ in *Figure 5—figure supplement 1aii* was isolated through subtraction of the $S_{ph}$ from the $V_{ph}$, before baseline subtraction.

## Immunohistochemistry

Immunohistochemical analysis of virus expression in combination with identification of 5-HT neurons via tryptophan-hydroxylase immunohistochemistry (mouse anti TPH antibody, dilution 1:200; Sigma-Aldrich with secondary antibody DyLight 488 donkey anti-mouse) was performed after four weeks of virus expression. Mouse brains were fixated for 2 h following perfusion with 4% paraformaldehyde in phosphate buffered saline (PBS, pH 7.4) and stored in cryoprotectant solution (30% sucrose in PBS). Coronal sections (30 μM) were collected in 24-well plates in tris-buffered saline (TBS, pH 7.5). Sections were rinsed three times in TBS and subsequently blocked with 0.1% TBST (TBS + Triton X-100) with 3% normal donkey serum (NDS, pH 7.2) for one hour at room temperature. The blocking serum

was aspirated, and sections were incubated overnight at 4°C on an orbital shaker with primary antibodies diluted in 1.5% NDS in 0.1% TBST. Brain slices were washed with TBS three times and incubated with anti–species-specific secondary antibodies (DyLight 488 donkey anti-mouse, 1:500) in 1.5% NDS in 0.1% TBST for one hour at room temperature. Sections were mounted onto Superfrost/Plus Microscope Slides (Thermo Scientific) and coverslipped using Roti-Mount FluorCare (Carl Roth).

Immunohistochemical analysis of virus expression in combination with cFos staining after optrode recordings in the DRN was conducted with rabbit anti cFos antibody (sc-52, dilution 1:1000, Santa Cruz Biotechnology) in combination with secondary antibody Alexa Fluor 488 donkey anti-rabbit, dilution 1:500, Life Technologies).

Digital images were acquired from brain sections using a Leica TCS SP5 confocal laser scanning microscope interfaced with a personal computer running Leica Application Suite AF 2.6 software. Objectives of 10×0.3 NA and 20×0.7 NA were used to capture images. Sequential Z-stacks were created for each section. Captured images were transferred into ImageJ 1.45 s (National Institutes of Health) for processing and image overlay.

## Data analysis

RCaMP signals were averaged across trials. In order to remove differences in noise levels due to spatial inhomogeneity in illumination, pixels were divided by the average of pre-stimulus activity (comprising 200 ms after the onset of recording). Furthermore, average blank conditions (see above) were subtracted from all conditions to remove non-neuronal signals related to heartbeat and breathing. Dividing the outcome to the mean blank signal leads to a unitless relative signal of fluorescence, denoted by $\Delta f/f$. In some cases, independent component analysis was used as a second pre-processing step to remove heartbeat and respiratory artifacts (*Maeda et al., 2001*; *Spors and Grinvald, 2002*), as well as the photostimulation artifact with a distinct 20 Hz frequency component. These steps were applied to each trial separately and then averaged. The resulting averaged signals were smoothed using a two-dimensional Gaussian filter (σ=20 pixels) and a high-pass Butterworth filter (order 4, cut-off frequency σ=33 pixels). All spatially averaged traces shown are averages across pixels in the selected ROI, and each response trace is normalized to the first peak in response to the first visual stimulus in control conditions. These data were then averaged across trials, experiments, and mice.

The amplitude of the evoked responses (*Figure 5bi*) is calculated as the average of activity within $w_1$ (400 ms time window around the peak of the visual response to each visual stimulus, *Figure 5ai*, inset) using normalized traces. The baseline component is calculated as the average of activity within $w_2$ (200 ms time window before the onset of each visual stimulus, *Figure 5ai*, inset). The magnitude of responses is calculated as the absolute change between amplitude and baseline values for each stimulus. The same approach is used for the MUA recordings shown in *Figure 5—figure supplement 1b*.

Latency and duration. Response latency was calculated as the point in time where responses reach the significance threshold, which is defined as the time when activity is 2xSD above mean of activity over 700 ms before stimulus onset. Response duration was calculated as the time during which activity levels are >50% of peak activity (red dots and the dark grey area in *Figure 8—figure supplement 1a*, $t_{50}$).

Contrast tuning function. Evoked responses to different visual contrasts were normalized to the peak of the evoked response at 100% grating contrast. Peak maxima of each contrast response (mean over experiments) were fitted using the Naka-Rushton function (*Naka and Rushton, 1966*):

$$\mathbf{R} = \mathbf{R_{max}} \frac{c^n}{(c^n + c_{50}{}^n)} + \mathbf{R_0} \tag{6}$$

in which $R_{max}$ is the maximum response, $c$ is the fractional contrast, $c_{50}$ is the contrast at which its response is half of the maximum response, n is proportional to the slope of the curve at $c_{50}$ and $R_0$ is the offset of responses R.

Linear regression. In order to test how the firing rates of neurons are scaled with photostimulation, we calculated average firing rates during pre- and post photostimulation time intervals. Using a linear regressor, *Equation 7*, the average firing rate in the post photostimulation time window was fitted:

$$fr_{post} = b_1 + b_2 \times ph + b_3 \times fr_{pre} + b_4 \times fr_{pre} \times ph \qquad (7)$$

Where $fr_{post}$ and $fr_{pre}$ are the average firing rates in the post and pre photostimulation windows, respectively. **Ph** stands for photostimulation and is equal to 0 for the control condition and 1 for conditions with photostimulation. Also, $b_1$ and $b_2$ form the intercepts and $b_3$ and $b_4$ form the slope of the line fitting to the data. Therefore $b_2$ and $b_4$ are the subtractive and divisive terms, respectively, accounting for suppression effects due to photostimulation. Firing rates of all recorded units are normalized by the maximum pre-firing rate. To avoid floor effects, units with a pre-firing rate of less than 1 Hz were excluded from analysis (using units with higher pre-firing rate produce even stronger suppression).

## Statistical analysis

In this study, one sided t test: one-sample, two-sample, and paired t test with permutation correction for multiple comparisons, are used to assess the significance of the values (mean ± SEM) if not stated otherwise.

# Additional information

## Funding

| Funder | Grant reference number | Author |
|---|---|---|
| Deutsche Forschungsgemeinschaft | JA 945/5-1 | Dirk Jancke |
| Deutsche Forschungsgemeinschaft | JA 945/4-1 | Dirk Jancke |
| Deutsche Forschungsgemeinschaft | HE 2471/12-1 | Stefan Herlitze |
| Deutsche Forschungsgemeinschaft | 2471/18-1 | Stefan Herlitze |
| Deutsche Forschungsgemeinschaft | Project ID 122679504 - SFB 874 (project part A2,D.J. and project part B10,S.H.) | Stefan Herlitze Dirk Jancke |
| Deutsche Forschungsgemeinschaft | JA 945/3-1 and SL 185/1-1, German-Israeli Project Cooperation (DIP) | Dirk Jancke |
| Deutsche Forschungsgemeinschaft | MA 5806/1-2 | Melanie D Mark |
| Deutsche Forschungsgemeinschaft | MA 5806/2-1 | Melanie D Mark |
| Deutsche Forschungsgemeinschaft | SFB 1280, DFG project no. 316803389 | Stefan Herlitze |

The funders had no role in study design, data collection and interpretation, or the decision to submit the work for publication.

## Author contributions

Zohre Azimi, Conceptualization, Data curation, Software, Formal analysis, Validation, Investigation, Visualization, Methodology, Writing - original draft, Writing - review and editing; Ruxandra Barzan, Validation, Investigation, Writing - review and editing, Data acquisition (electrophysiology); Katharina Spoida, Conceptualization, Resources, Visualization, Methodology, Data acquisition (DRN electrophysiology); Tatjana Surdin, Resources, Construction of AAV virus; In vitro validation of viral constructs; Patric Wollenweber, Resources, Methodology; Melanie D Mark, Resources, Supervision, Writing - review and editing; Stefan Herlitze, Conceptualization, Resources, Funding acquisition, Methodology, Project administration, Writing - review and editing; Dirk Jancke, Conceptualization, Resources, Data curation, Software, Formal analysis, Supervision, Funding acquisition, Validation,

Investigation, Visualization, Methodology, Writing - original draft, Project administration, Writing - review and editing

### Author ORCIDs
Zohre Azimi ⓘ https://orcid.org/0000-0001-5072-3264
Ruxandra Barzan ⓘ https://orcid.org/0000-0002-8021-9330
Stefan Herlitze ⓘ http://orcid.org/0000-0003-1785-0450
Dirk Jancke ⓘ https://orcid.org/0000-0001-8440-6259

### Ethics

Animal experimentation: All experimental procedures were carried out in accordance with the European Union Community Council guidelines and approved (Az.: 84-02.04.2014.A439) by the German Animal Care and Use Committee under the Deutsches Tierschutzgesetz and the NIH guidelines.

### Decision letter and Author response
Decision letter https://doi.org/10.7554/eLife.53552.sa1
Author response https://doi.org/10.7554/eLife.53552.sa2

## Additional files

### Supplementary files
• Transparent reporting form

### Data availability

All data generated or analysed during this study are included in the manuscript and supporting files. Source data file for Figure 3–9 are provided. Data can be downloaded from: https://doi.org/10.5061/dryad.931zcrjgk.

The following dataset was generated:

| Author(s) | Year | Dataset title | Dataset URL | Database and Identifier |
|---|---|---|---|---|
| Jancke D | 2020 | Separable gain control of ongoing and evoked activity in the visual cortex by serotonergic input | http://dx.doi.org/10.5061/dryad.931zcrjgk | Dryad Digital Repository, 10.5061/dryad.931zcrjgk |

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
