## [Decision Letter]

**Acceptance summary:**

This paper explores the effects of serotonergic photostimulation on the mouse visual cortex. By combining state-of-the-art optogenetics, wide-field optical imaging, electrophysiology and pharmacological manipulations in vivo, the authors show that serotonergic neurons modulate both spontaneous and visually evoked responses by two separable cellular mechanisms. This study highlights an important, yet less examined role of serotonin in sensory processing.

**Decision letter after peer review:**

[Editors’ note: the authors submitted for reconsideration following the decision after peer review. What follows is the decision letter after the first round of review.]

Thank you for submitting your work entitled "Subtraction and division of visual cortical population responses by the serotonergic system" for consideration by *eLife*. Your article has been reviewed by three peer reviewers, one of whom is a member of our Board of Reviewing Editors, and the evaluation has been overseen by a Senior Editor. The reviewers have opted to remain anonymous.

Our decision has been reached after consultation between the reviewers. Based on these discussions and the individual reviews below, we regret to inform you that your work will not be considered further for publication in *eLife*.

In coming to our decision, we were aware that you could make additional experiments that would address this issue, but those experiments are likely to require substantial time and effort. Therefore, as per *eLife*'s review policies, we decided to reject the manuscript. However, if you are able to collect single-unit data that support your claims, we would be willing to reconsider a resubmission. In that case, it would again be evaluated first by a Senior Editor and members of the Board of Reviewing Editors, who would decide whether or not to send it out for review.

*Reviewer #1:*

The neuromodulator serotonin is involved in a wide range of cognitive processes. Most current theories focus on its effects on learning, valuation and affect, leaving its involvement in sensory processing relatively less explored. In this timely report, Azimi et al. study this question by examining the effect of stimulating serotonergic neurons in the dorsal raphe nucleus on spontaneous and visually evoked activities in primary visual cortex of the mouse. They show that both are reduced following stimulation, but through different mechanisms: spontaneous activity is reduced due to activation of inhibitory 5-HT-1A receptors, and evoked activity is suppressed due to activation of excitatory 5-HT-2A receptors, suggesting a disynaptic circuit. From these data, the authors conclude that the serotonergic system provides gain control over the visual cortex that acts through both subtractive and divisive normalizations.

The experiments were very elegant, combining state-of-the-art optogenetic stimulation, wide-field optical imaging and pharmacological manipulations in vivo. While I generally support the publication of this paper, there are significant weaknesses in the manuscript, particularly in the statistical analysis and figure presentation that should be addressed before publication.

– The authors had made considerable efforts in performing detailed IHC, including testing expression of ChR2 in both DRN 5-HT neurons and their axons in V1, and looking at patterns of 5-HT receptor expression in V1. However, to support their claims, they should also provide quantification and statistical testing of these data.

– Generally speaking, the figures are very hard to understand and should be revised considerably. More specifically:

1) The authors should add figure legends for colored curves, shaded patches, rectangles on curves etc.

2) Axes titles should be more informative (for example, what is amplitude_W3_ in Figure 5C?) and panel headers will be helpful in cases where multiple panels present similar data (for example Figure 3A)

3) The authors should refrain from asking the reader to compare different figures (for example, comparing Figures 3A and 5B). If the claim is important, the comparison should be tested explicitly.

4) Figure 2, inhibition's delay in the example seems fairly long compared to the averages shown in Figure 3? It would be nice to see more detailed analysis of individual mouse effects (also showing individual mouse data, maybe as a supplementary figure).

5) It would also be nice to see schemes of experimental protocols, and, related to this, Figure 2 would benefit from an entire example session.

6) In Figure 4—figure supplement 1, it's hard to assess the effect of photostimulation on firing, perhaps adding PSTHs would help.

– Statistical analysis is lacking. The main text barely mentions which statistical tests were made (except for one test for the similarity of responses across cortical layers, which is not well described), what are the corresponding values being compared (in the form of mean ± SD, p values, n's etc.), and what are the justifications for choosing particular tests. The authors should add all this information, and also provide further details regarding the exact procedures for data normalization and averaging (for example, in Figure 5 "Values were averaged across time windows during photostimulation in which activity levels were significantly below baseline", sounds too arbitrary).

– In the Discussion, the authors only briefly mention the functional implications of their findings with respect to existing theories of 5-HT function, they should elaborate on this matter.

*Reviewer #2:*

Azimi et al. stimulated serotonergic neurons in the dorsal raphe nucleus while measuring activity in the primary visual cortex of anesthetized mice. They show that stimulation produces normalization of V1 responses, due to 5-HT1A and 5-HT2A receptors. This is an interesting set of experiments, that has important consequences for serotonin's function in cortex. Unfortunately, the data do not strongly support the conclusions.

1) My main concern regards the choice of widefield imaging to measure the effects of 5-HT stimulation in V1. Given the nonlinearities in the relationship between firing rates and RCaMP activity, couldn't subtraction (for example) be mistaken for normalization? Electrophysiological recordings are reported, but this is a small sample size, and it's unclear whether these are single-unit (as reported in the Materials and methods) or multi-unit (as reported in the main text) recordings. Related to this, it is not possible to determine the layer of the neurons recorded electrophysiologically, using the techniques reported.

2) There is a relatively sparse description of the methods. How were firing rates normalized? Why are some values > 1? I couldn't tell until reading the Materials and methods that mice were anesthetized.

3) 16 s of stimulation is quite long, and could have caused autoreceptor-mediated suppression of 5-HT neurons, producing the late excitatory effects in Figure 2. This could be discussed; ideally, the stimulation parameters should be manipulated in a more physiological range.

4) What is the baseline in Figure 6? Is it different for each contrast? Could it change by contrast (e.g. in Figure 2, it looks like there is a pre-stimulus decrease in activity)?

5) The references used to support statements in the Introduction and Discussion are peculiar. For example, some of the references in the third paragraph of the Introduction did not study sensory processing or use optogenetics. Why must the suppression observed here be mediated disynaptically (Discussion, second paragraph)? Wouldn't work from Gulledge and colleagues (Avesar and Gulldedge, 2012; Stephens, Avesar and Gulledge2014; Front Neural Circ 12, 2, 2018) suggest it could be done only using differential expression of different 5-HT receptors in pyramidal neurons?

*Reviewer #3:*

The manuscript by Azimi et al. Explores the effect of serotonergic photostimulation on the spontaneous activity and visual evoked responses in the visual cortex of mice. The authors perform a series of optogenetic experiments combined with bulk imaging and electrophysiology to show that by stimulating serotonergic neurons in the raphe nuclei both spontaneous and evoked responses are altered. Using pharmacological techniques they further show that the normalization of visual cortical activity by 5-HT PS consists of a subtractive suppression of spontaneous activity mediated by 5-HT1a receptors and a divisive suppression of response gain mediated by 5-HT2a receptors. In addition, I found the global cortical suppression by 5-HT PS very intriguing.

The topic of the manuscript is of broad interest, the experiments performed and data analysis sound, results intriguing and most conclusions drawn backed up by the data. Overall I think the manuscript is worthy of publication in a very good journal with a broad audience. I have several critiques, questions and observations and hope the authors take them in a constructive way. These do not lessen my overall support of the story.

1) The recordings of the present study were performed under anesthesia. This leads to two issues, first, the visual responses might be different (see Durand et al. 2016 JNeurosci for a comparison of awake vs. anesth visual responses) and second, photostimulation of 5-HT neurons in the anesthetized state might overestimate the effect of DRN photostimulation (in addition to the already unnaturally synchronous stimulation of neurons). Ideally a loss of function experiment in the awake state would nicely complement the results of this study, however I am aware that this might raise several technical concerns.

2) The authors have chosen to use ePet-cre mice, but ePet is not the most specific marker for 5-HT neurons (see two papers from Gabor Nyiri in Brain Struct Funct). This could be discussed in the manuscript, but the effects identified in the pharmacological experiments argue for specific serotonergic effects.

[Editors’ note: further revisions were suggested prior to acceptance, as described below.]

Thank you for submitting your article "Separable gain control of ongoing and evoked activity in the visual cortex by serotonergic input" for consideration by *eLife*. Your article has been reviewed by three peer reviewers, one of whom is a member of our Board of Reviewing Editors, and the evaluation has been overseen Joshua Gold as the Senior Editor. The following individual involved in review of your submission has agreed to reveal their identity: Magor Lorincz (Reviewer #3).

The reviewers have discussed the reviews with one another and the Reviewing Editor has drafted this decision to help you prepare a revised submission.

Summary:

Azimi et al. stimulated serotonergic neurons in the dorsal raphe nucleus while measuring activity in the primary visual cortex of mice. They show that stimulation produces normalization of V1 responses, due to 5-HT1A and 5-HT2A receptors. This is an interesting set of experiments, which have important consequences for serotonin's function in cortex. The careful revisions in light of the reviewers' comments and the added extracellular recordings, as well as experiments in awake animals, led to significant improvements in the manuscript's quality. As it stands, this manuscript is well written, interesting and timely. We do have the following comments.

Essential revisions:

1) Subsection “Distinct and independent contribution of 5-HT2A and 5-HT1A receptors to suppression of evoked and spontaneous activity”, first paragraph: The authors claim that the late increase in RCaMP signal that is usually observed following DRN photostimulation is suppressed by the application of MDL, yet do not support this claim with a statistical test. Furthermore, while the response does remain negative throughout drug application, it is increasing after an initial dip, and this increase cannot be explained by 5-HT1A receptor dynamics since these seem to be constant during photostimulation.

2) I do not understand the analysis that led to the conclusion that normalization is different between anesthetized and awake conditions. Specifically, if *V_ph_* / V is constant (as suggested by the analysis) then (*V_ph_* – *S_ph_*)/ V shouldn't be constant, and vice versa. This could be the case, however, if *S_ph_* is equal to zero. By looking at the traces shown in Figure 8Aii, it seems that at the time of peak evoked response, *S_ph_* is indeed close to 0. Thus, the difference between the two conditions may simply be the result of slower suppression of spontaneous activity or faster evoked responses in the awake states.

---

## [Author Response]

[Editors’ note: the authors resubmitted a revised version of the paper for consideration. What follows is the authors’ response to the first round of review.]

Reviewer #1:[…] The experiments were very elegant, combining state-of-the-art optogenetic stimulation, wide-field optical imaging and pharmacological manipulations in vivo. While I generally support the publication of this paper, there are significant weaknesses in the manuscript, particularly in the statistical analysis and figure presentation that should be addressed before publication.– The authors had made considerable efforts in performing detailed IHC, including testing expression of ChR2 in both DRN 5-HT neurons and their axons in V1, and looking at patterns of 5-HT receptor expression in V1. However, to support their claims, they should also provide quantification and statistical testing of these data.

The mentioned figure was meant as a proof of principle of the applied methodology rather than claiming quantifications. However, we agree with the reviewer that a thorough quantification would be necessary to support any specific claims about anatomical serotonergic innervation patterns in the visual cortex. As the focus of our study is on functional changes on V1 processing dynamics induced by activation of serotonergic neurons in the DRN and not on anatomical aspects (like projection pattern of fibers of 5-HT neurons in V1), we decided to skip this figure and refer to previous studies that have investigated such aspects in great detail (Dugué et al., 2014; Hale and Lowry, 2011; Leysen, 2004; Puig and Gulledge, 2011). However, we would like to keep the staining pattern of 5-HT neurons in the DRN expressing flx-ChR2 (former Figure 1C, now Figure 1B.) since it serves the depiction of experimental procedures.

– Generally speaking, the figures are very hard to understand and should be revised considerably. More specifically:1) The authors should add figure legends for colored curves, shaded patches, rectangles on curves etc.

Done – we added figure legends for all such elements.

2) Axes titles should be more informative (for example, what is amplitude_W3_ in Figure 5C?) and panel headers will be helpful in cases where multiple panels present similar data (for example Figure 3A).

Done – here applicable, axes titles were changed to exactly describe the presented parameter identities. Further, we added w of used abbreviations, and added panel headers (for instance, in Figure 5, former Figure 3, and Figures 6 and 7, several panel headers like “Visual stimulus”, “Amplitude”, “Magnitude”, etc. were added).

3) The authors should refrain from asking the reader to compare different figures (for example, comparing Figures 3A and 5B). If the claim is important, the comparison should be tested explicitly.

Done – To make the above-mentioned comparison easier to follow Figure 7 was added, which is summarizing the quantifications (magnitude and baseline values) shown in Figure 3 (spontaneous activity), Figure 5 (visually evoked activity) and Figure 6 (spontaneous and evoked activity with application of 5-HT1A and 5-HT2A receptors antagonists). Comparisons are verified by values and statistical tests in the main text.

4) Figure 2, inhibition's delay in the example seems fairly long compared to the averages shown in Figure 3? It would be nice to see more detailed analysis of individual mouse effects (also showing individual mouse data, maybe as a supplementary figure).

Done – Figure 2 depicts now all individual experiments. The difference in the onset of the suppression is calculated as mean ± SEM and stated in the main text. Average traces shown in Figure 3 and 5 were complemented with outlined SEM.

5) It would also be nice to see schemes of experimental protocols, and, related to this, Figure 2 would benefit from an entire example session.

Done – Figure 2A shows now all experimental protocols (systematically from top to bottom) and includes the time traces of activity recorded during the full time span of the trials. Also individual axes annotations and panel headers (depicting each stimulation condition) were added.

6) In Figure 4—figure supplement 1, it's hard to assess the effect of photostimulation on firing, perhaps adding PSTHs would help.

Done – PSTHs are added.

– Statistical analysis is lacking. The main text barely mentions which statistical tests were made (except for one test for the similarity of responses across cortical layers, which is not well described), what are the corresponding values being compared (in the form of mean ± SD, p values, n's etc.), and what are the justifications for choosing particular tests. The authors should add all this information, and also provide further details regarding the exact procedures for data normalization and averaging (for example, in Figure 5 "Values were averaged across time windows during photostimulation in which activity levels were significantly below baseline", sounds too arbitrary).

Done – whenever values mentioned in the main text the mean ± SEM is reported along with p values, type of the statistical test and number of subjects for that test. In addition, in the Materials and methods section the exact procedures of normalization, scaling or any other quantification steps are described. To find the onset of suppression and its effective duration (based on comparison between the traces of S and *S_ph_* with paired t test corrected for multiple comparisons), we chose a window based on significance value: “The suppression is significant 680 ms (n=8 animals, p=0.04; paired t test with permutation correction) after the onset of DRN photostimulation, …”. “By restricting the averaging window to the period where suppression of the RCaMP signal is significant (as marked with red rectangle in Figure 3Ai.), …”

– In the Discussion, the authors only briefly mention the functional implications of their findings with respect to existing theories of 5-HT function, they should elaborate on this matter.

Done – we rewrote the Discussion and also discussed most recent publications closely related to our findings.

We wrote: “In two recent studies, selective activation of 5-HT2A receptors produced a strikingly consistent suppressive effect on the gain of visually evoked population responses (Eickelbeck et al., 2019; Michaiel et al., 2019), despite cell-type- and layer-specific differences across single cells (Michaiel et al., 2019). […] Our study may help to develop new ways of diagnosis and therapy (Carhart-Harris et al., 2018) by rebalancing of these components.”

In the Introduction we added: “Intriguingly, using optogenetic stimulation of 5-HT neurons in the mouse DRN (Dugué et al., 2014; Kapoor et al., 2016; Matias et al., 2017), only one study in olfactory cortex has shown so far a 5-HT-induced (divisive) gain control of spontaneous firing, without any effect on the gain of stimulus-driven population responses (Lottem et al., 2016). This suggests that DRN activation could separately reduce the weight of ongoing cortical activity relative to evoked activity (Lottem et al., 2016), thereby possibly changing the balance of integration between internal priors (Berkes et al., 2011; Fiser et al., 2010) and external sensory input (Lottem et al., 2016).”

Reviewer #2:1) My main concern regards the choice of widefield imaging to measure the effects of 5-HT stimulation in V1. Given the nonlinearities in the relationship between firing rates and RCaMP activity, couldn't subtraction (for example) be mistaken for normalization? Electrophysiological recordings are reported, but this is a small sample size, and it's unclear whether these are single-unit (as reported in the Materials and methods) or multi-unit (as reported in the main text) recordings. Related to this, it is not possible to determine the layer of the neurons recorded electrophysiologically, using the techniques reported.

We thank the reviewer for her/his helpful review. We agree with the reviewer’s concern that, given potential nonlinearities in the relationship between firing rates and RCaMP activity, one should be careful in interpreting RCaMP signals with respect to spiking activity, which is, however, frequently ignored in recent studies.

New electrophysiological data added – We have performed a substantial number of new experiments using extracellular recordings of multi-unit activity and recorded firing rates based on well-isolated spikes in a new cohort of animals. The obtained extensive data set (previously 20 MUAs, new sample size in total 212 MUAs) fully supports our previous claims (please see new Figure 3, Figure 3—figure supplement 1, figure 4, Figure 5—figure supplement 2 and 3, Figure 8 and Figure 8—figure supplement 2). For both Ca^2+^ data and MUA data the 5-HT-induced suppression can be accounted for by a two-step normalization process (subtraction of spontaneous activity and scaling the gain of evoked activity, Figure 8di. and dii.). Therefore, potential nonlinearities between the RCaMP signal and firing rates do not affect our previous claims.

Corrected – we now refer to recording depth instead of layers. Thank you for the hint.

2) There is a relatively sparse description of the methods. How were firing rates normalized?

We generally improved explanations of the quantifications and analytical methods. In order to obtain a data format that allows comparison between firing rates and Ca^2+^ widefield imaging signals we employed two types of normalization:

“Traces of spike counts were averaged over time (200 ms bins) and across trials for each recorded condition. […] The normalized traces were then averaged over all the units and animals.”

With respect to contrast tuning we wrote: “Evoked responses to different visual contrasts were normalized to the peak of the evoked response at 100% grating contrast.”

Why are some values > 1?

As described above, all traces were divided by the average of activity obtained in response to the first presented visual stimulus (thus, acting as the reference response to all following stimuli). The mentioned later rise of the RCaMP signal that elevates evoked RCaMP signals to higher values than for the first stimulus, causes amplitude values >1.

I couldn't tell until reading the Materials and methods that mice were anesthetized.

Clarified – We now clarify that anesthetized animals were used directly in the Abstract, as well as in Figure 1 legend, Introduction, Results, and in Materials and methods. We further have now included new experiments where we compare results obtained under anesthesia with results obtained in awake animals (Figures 8 and 9). In this context we phrase: “Hence, the impact of activating the DRN on cortical activity may differ under wakeful conditions compared to conditions where the animals are anesthetized.”

3) 16 s of stimulation is quite long, and could have caused autoreceptor-mediated suppression of 5-HT neurons, producing the late excitatory effects in Figure 2. This could be discussed; ideally, the stimulation parameters should be manipulated in a more physiological range.

Thank you for raising these important points, which we have now addressed in detail in the Discussion.

We think that autoreceptor-mediated suppression is an unlikely mechanism contributing to our results for the following reasons: “Autoreceptor-mediated suppression of 5-HT neurons within the DRN is an unlikely explanation for producing such late increase. […] Second, we show using MUA that spiking activity in V1 is consistently suppressed during photostimulation. Third, the evoked component of visual responses remains suppressed during DRN photostimulation and finally, the late increase of RCaMP signals was selectively blocked by 5-HT2A antagonist.”

With respect to the late excitatory effect we wrote: “We suggest that intracellular Ca^2+^-accumulation of cortical neurons produces the late rise in the RCaMP signal (Eickelbeck et al., 2019) due to 5-HT receptor-mediated activation of the Gq/11 pathway (Jang et al., 2012; Millan et al., 2008) with further activation of store-operated channels (Celada et al., 2013). Finally, the late increase of the RCaMP signal did not affect our conclusions but may be considered when using wide-field Ca^2+^ recordings of cortical activity in future studies.”

We think that the used stimulation parameters are in a physiological range, as much as an experimental manipulation can be per se, in order to derive reasonable effect sizes. However, please note that even though we may drive neurons in the DRN with 20 Hz photostimulation strongly, it was recently shown that units in the intact DRN never fire at or close to 20 Hz in vivo (Grandjean et al., 2019). In this study it is also shown that DRN neurons increase their firing throughout 20 s of 20 Hz photostimulation without adaptation. Altogether, this indicates that DRN neurons may fire with their maximal output but within their physiological limits.

We wrote: “The DRN was photostimulated for 16 s in the first set of our experiments. This is relatively long in comparison to the brisk activation of DRN neurons observed in certain behavioral contexts (Heym et al., 1982; Ranade and Mainen, 2009). However, this does not exclude additional longer-lasting tonic increase in DRN activity under natural conditions. For instance, direct retinal projections to the DRN (Pickard et al., 2015; Ren et al., 2013; Zhang et al., 2016) suggest its regulation well beyond the stimulation time window chosen in our study. In addition, slow periodic changes in 5-HT release occur naturally during day-night cycle (Portas et al., 2000; Tyree and de Lecea, 2017) and depend on seasonal factors (Spindelegger et al., 2012). Importantly, using the ePet-Cre mouse model and comparable photostimulation parameters as in our experiments (i.e. similar blue light intensity and frequency) to activate the DRN via fiber optics, a consistent increase of DRN activity in fMRI and MUA recordings over 20 s of photostimulation was revealed (Grandjean et al., 2019). Prolonged (12s) photostimulation of the DRN in SERT-Cre mice increased perceptual thresholds for tactile stimuli (Dugué et al., 2014), altogether suggesting that sustained or repeated elevation of DRN activity correlates with meaningful and behaviorally relevant changes in neuronal responses (Correia et al., 2017; Lőrincz and Adamantidis, 2017; Miyazaki et al., 2011). “

4) What is the baseline in Figure 6? Is it different for each contrast?

In the revised manuscript the corresponding figure is Figure 8. The baseline that is subtracted from evoked responses to each contrast is derived from the condition that records ongoing activity during photostimulation (*S_ph_*, Figure 8aii., light blue trace). Thus, this baseline is recorded once and is independent of contrast.

Could it change by contrast (e.g. in Figure 2, it looks like there is a pre-stimulus decrease in activity)?

Based on our results, the 5-HT-induced changes in spontaneous activity are proportional to pre-photostimulation firing rates and can be explained by a divisive gain control independent of visual input (Figure 4 and Figure 5—figure supplement 3). Following the linear summation hypothesis, ongoing activity and evoked responses are independent (Arieli, Sterkin, Grinvald, and Aertsen, 1996; Deneux and Grinvald, 2016; Ferezou and Deneux, 2017; Ferezou et al., 2007) and therefore, baseline (ongoing) activity should not be differently affected by different contrasts. Indeed, when using a two-step normalization process that treats these components as independent, we were able to recover the contrast tuning function of control conditions from those under the influence of 5-HT (Figure 8di/ii and Figure 9cii).

Nonetheless, we are aware of recent studies that consider also reciprocal effects of external stimuli on the structure of ongoing activity (Deneux and Grinvald, 2016; Ferezou and Deneux, 2017; He, 2013; Sadaghiani, Hesselmann, Friston, and Kleinschmidt, 2010). However, we felt that discussing this issue would go too far beyond our current findings.

Regarding the part of the reviewer's question in brackets: The pre-stimulus decrease seen in this example reflects the variability in ongoing activity. Please see the newly included other examples for comparison (Figure 2a). This pre-stimulus variability is negligible in comparison to the suppression induced by 5-HT input (Figure 2b, *S_ph_* condition and Figure 3).

5) The references used to support statements in the Introduction and Discussion are peculiar. For example, some of the references in the third paragraph of the Introduction did not study sensory processing or use optogenetics.

Corrected – statements in Introduction and Discussion are now supplemented with adequate references, separating those that used optogenetics and those that did not.

Why must the suppression observed here be mediated disynaptically (Discussion, second paragraph)? Wouldn't work from Gulledge and colleagues (Avesar and Gulldedge, 2012; Stephens, Avesar and Gulledge, 2014; Front Neural Circ 12, 2, 2018) suggest it could be done only using differential expression of different 5-HT receptors in pyramidal neurons?

Possible misunderstanding – we did not suggest that all of the observed suppression was mediated disynaptically. We propose that a major part of the suppression, contributing to controlling response gain, involves activation of 5-HT2A receptors. We suggest that part of this suppression (note that this suppression was blocked by the 5-HT2A antagonist) is mediated via interneurons that are activated via 5-HT2A. Importantly, we mentioned another possible monosynaptic mechanism possibly contributing to suppression of response gain, that is, a 5-HT2A-mediated shunting in pyramidal neurons.

We skipped the terms mono- and di-synaptic, however, to improve clarity.

References added – further, we refer to two of the references suggested by the reviewer in the Discussion: “Thus, reduction in response duration under the influence of 5-HT may indicate involvement of shunting inhibition in pyramidal neurons to effectively control response gain. Suprathreshold excitatory drive by 5-HT2A activation may further be amplified by shaping the duration of 5-HT1A-mediated inhibition (Avesar and Gulledge, 2012; Stephens et al., 2014).”

Reviewer #3:[…] The topic of the manuscript is of broad interest, the experiments performed and data analysis sound, results intriguing and most conclusions drawn backed up by the data. Overall I think the manuscript is worthy of publication in a very good journal with a broad audience. I have several critiques, questions and observations and hope the authors take them in a constructive way. These do not lessen my overall support of the story.1) The recordings of the present study were performed under anesthesia. This leads to two issues, first, the visual responses might be different (see Durand et al. 2016 JNeurosci for a comparison of awake vs. anesth visual responses) and second, photostimulation of 5-HT neurons in the anesthetized state might overestimate the effect of DRN photostimulation (in addition to the already unnaturally synchronous stimulation of neurons). Ideally a loss of function experiment in the awake state would nicely complement the results of this study, however I am aware that this might raise several technical concerns.

New experiments added under awake conditions – We thank the reviewer for her/his comments. Motivated by these comments, we performed a new row of experiments, in which we imaged animals in the awake state. We are most happy that these new data further substantiated and elaborated our major previous claims (please see new Figure 9).

We added: “Finally, we explored the extent to which the above conclusion depends on the cortical state. […] Therefore, we recorded cortical activity of awake mice (n=5) that were head-fixed and habituated to walk and stay on a treadmill (using the same experimental paradigm as shown in Figure 8).”

We concluded a comparison between awake and anaesthetized conditions as follows: “We found that normalization of responses in the anesthetized state is achieved with additive contribution of ongoing activity, indicating integration of spontaneous and evoked activity with increased weight of internal priors. In awake mice, normalization is independent of suppression in ongoing activity. This suggests that 5-HT provides a discrete divisive gain control of spontaneous ongoing and evoked visual activity, while preserving information content and regulating the balance between these components depending on the cortical state.”

We agree with the reviewer that a loss of function experiment in the awake state would nicely complement the results, but in view of the many results obtained in our study and the extensive additional data that would be required, we feel that this would go beyond what is feasible in a single study. In addition to the technical difficulties pointed out by the reviewer, it will also be challenging to realize inhibition of DRN neurons within physiological ranges. Certainly, we will follow up on this difficult issue in our future studies.

2) The authors have chosen to use ePet-cre mice, but ePet is not the most specific marker for 5-HT neurons (see two papers from Gabor Nyiri in Brain Struct Funct). This could be discussed in the manuscript, but the effects identified in the pharmacological experiments argue for specific serotonergic effects.

We thank the reviewer for this important remark. In the revised Discussion section we have addressed this concern. We wrote: “It should also be noted that besides 5-HT neurons, subpopulations of glutamatergic cells in the raphe nuclei express ePet (McDevitt et al., 2014; Sos et al., 2017). […] However, as seen in our experiments pharmacological blocking of 5-HT receptors in V1 largely diminished the induced suppressive effects, contribution from other modulatory systems appears negligible for the present study.”

[Editors’ note: what follows is the authors’ response to the second round of review.]

Essential revisions:1) Subsection “Distinct and independent contribution of 5-HT2A and 5-HT1A receptors to suppression of evoked and spontaneous activity”, first paragraph: The authors claim that the late increase in RCaMP signal that is usually observed following DRN photostimulation is suppressed by the application of MDL, yet do not support this claim with a statistical test.

We removed the qualitative statement and report instead the p-value of a statistical test that compares baseline suppression of controls during DRN photostimulation (*S_ph_*) with *S_ph_* suppression obtained during application of MDL. Specifically, we analyzed the questioned rising part of the RCaMP signal after the initial dip. We found that the rising part of spontaneous activity during MDL application was significantly lower in comparison to controls (p=0.04, two-sample t test). This supports our suggestion of a reduction of intracellular Ca^2+^ accumulation when blocking 5-HT2A receptors and confirms that the late increase in RCaMP signal is partially suppressed by application of MDL.

Furthermore, while the response does remain negative throughout drug application, it is increasing after an initial dip, and this increase cannot be explained by 5-HT1A receptor dynamics since these seem to be constant during photostimulation.

We did not intend to claim that the late increase in the RCaMP signal is completely suppressed or is solely based on 5-HT2A receptor activation. The rising RCaMP signal after the initial dip during 5 HT2A blocking clearly suggests a residual contribution of other 5-HT receptors involved in intracellular Ca^2+^ accumulation that remain active during MDL application. We thank the reviewers for raising this important point that we now have addressed to avoid misunderstanding. As noted above, we removed the previous statement that the RCaMP signal remained negative throughout drug application and added further information about a likely contribution of other 5-HT receptors involved in the rise of the RCaMP signal, both in Results and in the Discussion.

2) I do not understand the analysis that led to the conclusion that normalization is different between anesthetized and awake conditions. Specifically, if V_ph_ / V is constant (as suggested by the analysis) then (V_ph_ – S_ph_)/ V shouldn't be constant, and vice versa. This could be the case, however, if S_ph_ is equal to zero. By looking at the traces shown in Figure 8Aii, it seems that at the time of peak evoked response, S_ph_ is indeed close to 0. Thus, the difference between the two conditions may simply be the result of slower suppression of spontaneous activity or faster evoked responses in the awake states.

We thank the reviewer for this important point, which we address here in more detail. In the manuscript we added a new section about the contribution of spontaneous activity to normalization together with a new supplementary figure (Figure 9—figure supplement 2). To answer this question one needs to consider the suppressive gain of the evoked visual response as an adjusting factor, which we show is different in anesthetized and awake conditions. As suggested, assuming that V_ph_/V is constant, the control visual response (V) can be approximated by the 5-HT-induced gain (i.e. the inverse of c) and the visual response during photostimulation (V_ph_):

V_ph_/V=c→V=1/c×V_ph_ (1)

Our main hypothesis here is that V_ph_ is a linear combination of the evoked component (E_ph_) and its baseline component (b_ph_). Further, we show (Figure 5b.) that b_ph_ is approximately equal to the suppression in spontaneous activity (S_ph_) and that the effect of photostimulation is always suppressive (i.e. g>1).

V_ph_=E_ph_+b_ph_

b_ph_≅S_ph_ (2)

1/c=g→c<1;g>1

Thus, under these considerations Equation 1 can be rewritten as:

V=g(E_ph_+S_ph_) (3)

V=g×E_ph_+g×S_ph_ (4).

Equation 4, shows that the gain (g) affects both the evoked and the spontaneous component and that normalization depends on their relative weights (see new Figure 9—figure supplement 2.).

Figure 9—figure supplement 2ai and aii. demonstrate how normalization influences the relative contribution of the baseline component (g×S_ph_). In the anesthetized state, the weight of the baseline component is significantly larger than in the awake condition (-0.36±0.11, n=18, -0.11±0.02, n=5; p<0.001, two-sample t test). This affects normalization particularly at low contrasts, where the weights of the baseline and evoked components are comparable (Figure 9—figure supplement 2bi). This further suggests that the baseline component in the anesthetized state is a likely candidate for an additive contribution to normalization. In contrast, in the awake state, the weight of the components in Equation 4 is significantly (p<0.01, one-sample t test) biased towards the evoked component (Figure 9—figure supplement 2bii.)

One might argue, however, that the reduced contribution of the baseline component in the awake state is merely due to its smaller value (i.e. S_ph aw_≤S_ph an_) at the time of the peak of the evoked response (t_pk_). Indeed, as the reviewer noticed correctly, we find that V_ph_ in the awake state reaches peak intensities significantly earlier than in the anesthetized state (mean over all contrasts: 311±32 ms (n=5 awake mice), 441±41 ms (n=18 anesthetized mice), p<0.001 two-sample t test). Consequently, the amount of suppression in S_ph_ at t_pk_ in the awake state, although significant (-0.09±0.03, n=5, p<0.01, one-sample t test), is lower as compared to the anesthetized state (-0.17±0.03, n=18, p<0.001, two-sample t test). To estimate the sensitivity of normalization to changes in the value of the baseline component in the awake state, we considered the same value of S_ph_ at t_pk_ as found for the anesthetized state (-0.17, i.e. S_ph aw_≅S_ph an_), while all other parameters in Equation 4 were kept the same (note that values of the evoked component do not change during photostimulation, Figure 4biii.). We find that even though the relative weight of the baseline component increases (-0.22±0.03, n=5, p<0.01, one-sample t test), its weight is still significantly smaller than in the anesthetized state (p<0.001 two-sample t test) and also significantly smaller than the weight of the evoked component over all contrasts (Figure 9—figure supplement 2biii.).

S_ph aw_≤S_ph an_ and g_aw_<g_as_ (5)

Our data suggest a reduction in the overall gain as the major difference between the awake and the anesthetized state (g_aw_<g_an_), as indicated by significantly decreased suppression of the peak response in the awake (0.75±0.11) compared to the anesthetized state (0.50±0.17, p<0.01 two-sample t test). As implicated by the calculations above and considering Equation 5 a lower gain in the awake state significantly decreases the relative weight of the baseline component [(g×S_ph_)_aw_ <(g×S_ph_)_an_]. Additionally, as shown in our data, given that for all contrasts (g×E_ph_) is significantly higher than (g×S_ph_) in the awake state, Equation 4 can be written as

V≅g×E_ph_ (6).

Consequently, normalization is largely controlled by the gain of the evoked component. This supports our conclusion that in the awake state, through a reduction in the 5-HT-induced gain, the weight of contribution to normalization is biased towards the evoked component and largely achieved without the requirement of a significant additive contribution of the baseline component, as is the case in the anesthetized state (cf. Figures 8di-ii and Figure 9ci and cii).